# SMAC: Score-Matched Actor-Critics for Robust Offline-to-Online Transfer

**Nathan S. de Lara** [1 2]  **Florian Shkurti** [1 2]

## Abstract

Modern offline Reinforcement Learning (RL) methods can learn performant actor-critics, but fine-tuning them online with value-based RL often causes an immediate performance drop. We provide evidence consistent with a geometric explanation: prior offline methods converge to high-reward solutions separated from online optima by low-reward valleys that gradient-based fine-tuning traverses. We introduce Score Matched Actor-Critic (SMAC), an offline RL method that learns actor-critics more compatible with subsequent online fine-tuning by regularizing the Q-function's action-gradient toward the score of the dataset action distribution. SMAC converges to offline maxima connected to better online maxima by monotonically improving reward paths, and transfers smoothly to Soft Actor-Critic and TD3 in 6/6 D4RL tasks. In 4/6 environments, SMAC reduces regret by 34–58% relative to the best baseline.

## 1. Introduction

Fine-tuning actor-critics from offline RL checkpoints with online value-based methods often triggers an immediate drop in performance. This path matters: when online interaction is expensive, time-limited, or physically deployed, an early collapse can erase the practical benefit of offline pre-training even if the policy eventually recovers. More broadly, offline RL should provide reusable actor-critic initializations that can be paired with data-efficient online algorithms, rather than checkpoints that only work when fine-tuned with the same regularized objective that produced them. For offline RL to support such a pre-train fine-tune paradigm, it must produce checkpoints that are not only strong before interaction begins, but also compatible with the online updates that follow. Standard online actor-critic

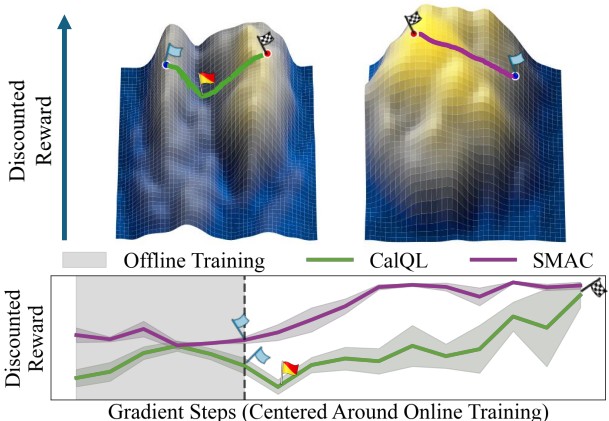

*Figure 1.* **Past offline RL methods converge to maxima separated from online optima by low-reward valleys**. Top: reward landscapes on the Kitchen task for CalQL (left) and SMAC (right). Blue and checkered flags being the real locations of the pre-trained and fine-tuned checkpoints on the landscape respectively. The paths and red/yellow flag are illustrative annotations showing the hypothesized trajectory during transfer. Paths demonstrate the existence of a low reward valley between pre-trained and fine-tuned checkpoints when using CalQL. Our method SMAC has no such valleys and is on a unified hill with the fine-tuning checkpoint. Bottom: SMAC vs. CalQL performance in the Kitchen task. See Section 5 for analysis.

updates should be able to improve the checkpoint without first traversing low-reward regions of parameter space.

We study this compatibility through the geometry of the expected-return landscape. Our hypothesis is that prior offline RL methods often converge to high-reward checkpoints that are poorly connected to the optima reached by online actor-critic fine-tuning. In this case, gradient-based updates can pass through low-reward regions before recovering, producing the characteristic initial transfer drop. We formalize this using linear connectivity: two checkpoints are linearly connected if reward changes monotonically along the straight line between them in parameter space. In Section 5, we show that common offline RL methods are not linearly connected to the checkpoints found by online SAC fine-tuning in tasks where they suffer an initial drop.

Following this, we introduce Score Matched Actor-Critic (SMAC), an offline RL method designed to produce actor-critic checkpoints that online value-based methods can continue improving. SMAC extends SAC with a critic regular-

---

[1]Department of Computer Science, University of Toronto, Toronto, Canada [2]Vector Institute. Correspondence to: Nathan S. de Lara <nathan.delara@mail.utoronto.ca>.

*Proceedings of the $43^{rd}$ International Conference on Machine Learning*, Seoul, South Korea. PMLR 306, 2026. Copyright 2026 by the author(s).

izer, motivated by maximum-entropy RL (Haarnoja et al., 2017), that aligns the critic action-gradient $\nabla_a Q(s, a)$ with an estimate of the dataset action score $\nabla_a \log \pi^{\mathcal{D}}(a|s)$. This regularizer directly targets the geometry problem: it biases the offline critic toward local action-gradient structure that online SAC optima satisfy, making the resulting actor-critic more compatible with subsequent online updates. We also use Muon (Jordan, 2024) as an empirical stabilizer for the offline checkpoint.

Across our benchmarks, SMAC converges to offline maxima that are connected to online SAC maxima, supporting the view that connectivity explains offline-to-online transfer. SMAC transfers smoothly to SAC, TD3 (Fujimoto et al., 2018), and TD3+BC (Fujimoto & Gu, 2021). In 4/6 tasks, SMAC pre-training followed by SAC fine-tuning reduces regret by 34–58% relative to the best baseline.

Our contributions are: (i) evidence that offline-to-online performance drops coincide with poor linear connectivity between offline and online checkpoints; and (ii) SMAC, an offline actor-critic method designed to produce checkpoints that transfer smoothly to data-efficient online fine-tuning.

## 2. Preliminaries

We use the standard Markov Decision Process notation: states $s$, actions $a$, rewards $r$, policies $\pi(a|s)$, Q-functions $Q(s, a)$, expected return $\mathcal{J}(\pi)$, and offline datasets $\mathcal{D} = \{(s_i, a_i, r_i, s'_i)\}_{i=1}^N$. Background on RL, value-based RL, and offline RL is provided in Appendices A, B, and C.

**Offline-to-online transfer.** Fine-tuning an offline actor-critic online can push the policy into states and actions that were poorly covered by the offline data. In these regions, critic errors are difficult to correct before they influence the actor, so the first online updates can briefly reduce performance (Zhou et al., 2024; Nakamoto et al., 2023). Existing approaches usually handle this in one of two ways: they regularize the critic throughout both the offline and online phases (Nakamoto et al., 2023; Wen et al., 2024), or they add online policy regularization so the actor stays close to the data while it adapts (Dong et al., 2025a; Zhang et al., 2023a). TD3+BC (Fujimoto & Gu, 2021) is a simple representative of the second approach: it can stabilize fine-tuning, but often trades off adaptation speed and final performance.

**Local connectivity.** We study transfer through the geometry of expected return $\mathcal{J}(\pi)$ as a function of policy parameters. Two high-performing checkpoints are linearly connected if reward changes monotonically along the straight line between them in parameter space. This differs slightly from the standard supervised-learning notion of mode connectivity, where one typically studies low-loss paths between minima (Garipov et al., 2018; Frankle et al., 2020). The intuition is similar: if two solutions are connected by a high-

performing path, local optimization can move between them without crossing a low-reward region. If they are separated by a low-reward valley, fine-tuning may suffer an initial performance collapse even when both endpoints are good policies.

**Diffusion score estimates.** Diffusion models estimate scores $\nabla_x \log p(x)$ from samples by learning to reverse a noising process. For behavior-cloned diffusion policies, the least-noised prediction approximates $\nabla_a \log \pi^{\mathcal{D}}(a|s)$ (Ho et al., 2020; Song & Ermon, 2020). SMAC uses Reinforcement via Supervision (RvS) conditioning so the score estimate targets high-return actions rather than the average action under the dataset (Piche et al., 2022; Emmons et al., 2022). Diffusion training details are in Appendix O.

**Maximum-entropy identity.** Maximum-entropy RL optimizes reward together with an entropy bonus, encouraging policies that are high-return while remaining stochastic. At an optimum of this objective, the policy has Boltzmann form:

$$\log \pi^*(a|s) = \frac{1}{\alpha}Q^*(s, a) - \log \int_{\bar{a}} \exp\left(\frac{1}{\alpha}Q^*(s, \bar{a})\right) d\bar{a}.$$

Taking the gradient with respect to $a$ removes the normalizer and gives

$$\nabla_a \log \pi^*(a|s) = \frac{1}{\alpha}\nabla_a Q^*(s, a) \tag{1}$$

Thus, SAC-style optima align the policy score with the critic action-gradient. SMAC uses this identity as motivation for an offline regularizer, not as an assumption that the dataset policy is itself optimal.

## 3. Problem statement

Let $D$ be an offline dataset collected in an MDP $M$. An offline RL algorithm $A$ outputs an actor-critic initialization $(Q_0, \pi_0) = A(D)$. We fine-tune $(Q_0, \pi_0)$ with an online RL algorithm $F$ that iteratively produces $(Q_{t+1}, \pi_{t+1}) = F(Q_t, \pi_t)$ for $t = 1, \ldots, N-1$ using interaction data from $M$ and $\pi_t$. We evaluate $A$ by how well it supports fine-tuning across a family $\mathcal{F}$ of online RL algorithms using the following criteria:

1. **Stable transfer:** $\mathbb{E}[\mathcal{J}(\pi_1)] \geq \mathbb{E}[\mathcal{J}(\pi_0)]$ where the expectation is over inherent noise in algorithms. This corresponds to not having a drop during the initial phase of online fine-tuning.

2. **Low online regret:** $\text{Regret}_N := \sum_{t=1}^N (\mathcal{J}(\pi^*) - \mathcal{J}(\pi_t))$, where $\pi^* \in \arg\max_\pi \mathcal{J}(\pi)$.

The experiments reflect this setting by pre-training actor-critics offline, warm-starting an online replay buffer with

data from the offline policy, and then fine-tuning with online actor-critic updates.

SMAC addresses these criteria by optimizing the offline critic toward functions that satisfy a structural property of online SAC optima. Maximum-entropy RL motivates this target: at a SAC optimum, the policy score and critic action-gradient are aligned. SMAC encourages this alignment offline by regularizing the critic action-gradient toward an estimate of the dataset action score, giving the offline actor-critic local structure more compatible with later online updates. We define the regularizer in Section 6 and evaluate transfer in Section 7.

## 4. Experimental setup

We evaluate offline-to-online transfer using three measurements: online learning curves for transfer stability, online regret for the cost of poor transfer, and interpolation plots for linear connectivity between offline and fine-tuned checkpoints. Unless otherwise noted, each offline-to-online pair is evaluated with 5 seeds per environment; Figure 3 uses 4 seeds.

We compare against CalQL/CQL (Nakamoto et al., 2023; Kumar et al., 2020), IQL (Kostrikov et al., 2022), and TD3+BC (Fujimoto & Gu, 2021), since each returns actor-critics that can be fine-tuned by standard online actor-critic algorithms. We use CalQL when Monte-Carlo returns are available and CQL otherwise; details are in Appendices S.3 and Q. We evaluate on six D4RL (Fu et al., 2021) benchmarks: `hopper-medium-replay-v2`, `walker2d-medium-replay-v2`, `kitchen-partial-v0`, `door-binary-v0`, `pen-binary-v0`, and `relocate-binary-v0`; benchmark details are in Appendix S.1.

After offline training, we collect 5,000 on-policy examples from the offline policy to warm-start the replay buffer, following (Zhou et al., 2024; Nakamoto et al., 2023). During online fine-tuning, each update samples batches with 50% offline transitions and 50% online replay transitions. We train each online phase for 200,000 environment steps.

## 5. Linear connectivity of offline and online maxima

This section links the absence of linearly connected offline and online maxima to the instability often seen when fine-tuning offline actor–critics. We use reward landscapes to visualize reward geometry along fine-tuning directions. Reward landscapes are the RL analog to loss landscapes in supervised learning. We focus on SAC, as it is a representative online actor-critic algorithm. We plot reward landscapes in the `kitchen-partial-v0` environment. Offline RL algorithms achieve strong but suboptimal performance in

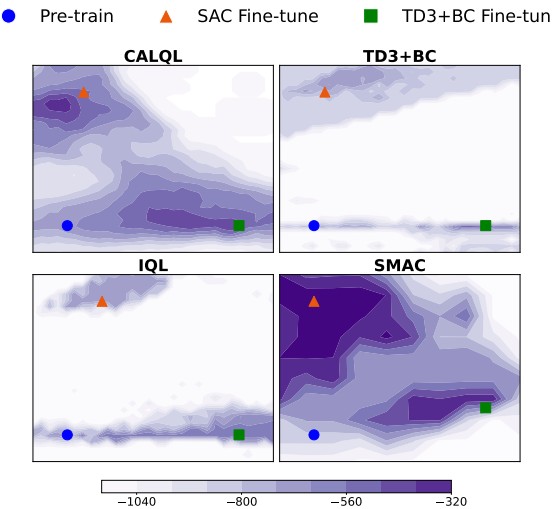

*Figure 2.* **Reward visualized along a plane in parameter space reveals difference in maxima found by different pre-training and fine-tuning methods on Kitchen task**. We see that the SAC maxima are wider and *not connected* to the pre-trained checkpoint along monotonically improving line across all baselines. Conversely, SAC maxima and SMAC maxima are linearly connected. Subplot titles denote offline algorithm used

the environment, so online fine-tuning can strongly improve, or worsen a good policy. We investigate the reward landscapes for offline RL baselines. We also show results for SMAC, the algorithm we present in Section 6 as a reference for effective transfer.

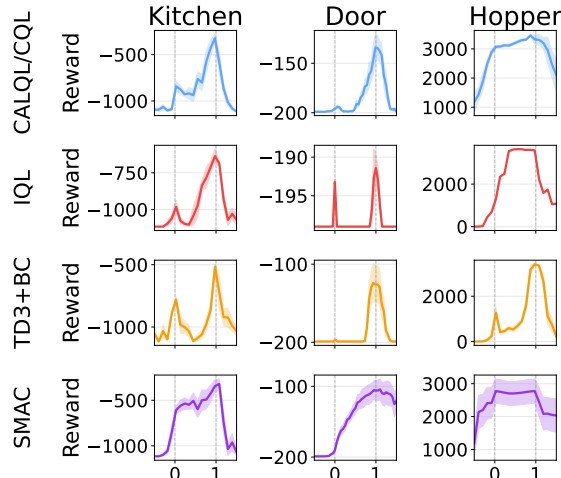

*Figure 3.* **Reward valleys when linearly interpolating between pre-training and fine-tuning checkpoints for all baselines in tasks show linearly disconnected maxima consistent with offline-to-online transfer performance in later plots**. We plot the performance along the line between the pre-trained checkpoint and final fine-tuning checkpoint for methods in kitchen-partial (left), door-binary (centre), and hopper-medium-replay (right). 0 is the pre-trained checkpoint, and 1 is the SAC fine-tuned checkpoint. Lines show mean over 4 seeds with shading being standard error.

In Figure 3, we interpolate between the pre-trained and fine-tuned checkpoints for different offline RL algorithms and

plot the average reward at different points along the line. More specifically, we pre-train to get parameters $\theta_{\text{offline}}$, and then fine-tune with SAC to get $\theta_{\text{online}}$. We plot performance on the line $\theta(t) = \theta_{\text{offline}} t + (1-t)\theta_{\text{online}}$ by rolling out with parameters $\theta(t)$ at various values of $t$. We observe that in the two left columns all algorithms except SMAC (ours) exhibit a drop in performance between $0$ (pre-trained checkpoint) and $1$ (fine-tuned checkpoint). Similarly, in Figure 5, all methods except SMAC suffer a drop in performance when transferring from offline-to-online training in the corresponding tasks. This suggests that in an environment where a drop in performance is observed upon transfer, the maxima for offline and online actor-critic methods do not lie on a concave subspace. In the `hopper-medium-replay-v2` environment, $3/4$ methods show that performance monotonically improves as you interpolate between checkpoints, confirming the results in Figure 5, where the corresponding $3/4$ methods transfer with no drop in performance.

As described in Section 2, TD3+BC, an offline algorithm, can fine-tune pre-trained actor-critics without suffering a dip in performance, but, at the cost of converging to suboptimal policies. A natural conclusion is that fine-tuning with TD3+BC moves the parameters in different directions that fine-tuning with SAC. In Figure 2, we visualize the loss landscape along these directions by looking at the plane defined by three parameterizations: a $\theta$ pre-trained by each subplots labelled algorithm, a SAC fine-tuned $\theta_1$, and a TD3+BC fine-tuned $\theta_2$. Both $\theta_1$ and $\theta_2$ are found by fine-tuning $\theta$ with the respective algorithms. The plane is spanned by $u := \theta_1 - \theta$, and $v := \theta_2 - \theta$, with performance visualized as a contour graph. We denote where in the plane the three parameters lie. We observe that the line between the pre-trained and SAC fine-tuned parameters travels through a low reward valley for all algorithms except SMAC. For all methods, the line between the pre-trained parameters and TD3+BC fine-tuned parameters is thin and high reward. Furthermore, we see that the parameters found by fine-tuning with SAC vs. TD3+BC are not linearly connected. In Appendix R, we describe how we visualize the planes.

We do acknowledge that optimization trajectories are unlikely to be exactly linear. It is possible that during optimization the parameters follow a curve outside the plotted plane. In Figure 4, we show the result of our attempt to check whether a projection of the parameters into 2D aligns with our observations in Figure 2. The plots show the t-SNE (van der Maaten & Hinton, 2008) projection of checkpoints along the pre-training and two fine-tuning optimization trajectories from Figure 2. While t-SNE cannot characterize global geometry, the projection is consistent with our planar visualization, in that SAC fine-tuning trajectories pass through regions of substantially lower reward before converging. This supports the interpretation that the performance collapse is associated with low-reward regions

between offline and online maxima.

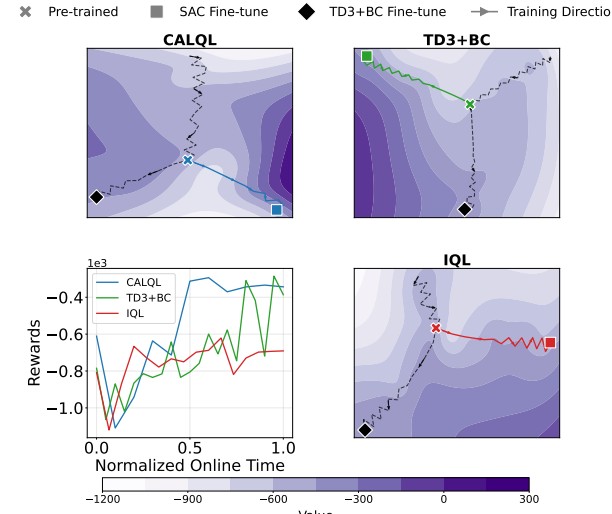

*Figure 4.* **t-SNE projections of training trajectories show linearly disconnected maxima**. We take the pre-training checkpoints, SAC fine-tuning checkpoints, and TD3+BC fine-tuning checkpoints and plot their T-SNE projections with lines and arrows signifying the training trajectory/ordering of the checkpoints. We observe that the projected checkpoints (i) travel in straight lines, and (ii) cross a valley of low reward when fine-tuned with SAC but not when fine-tuned with TD3+BC, providing evidence consistent with the reward valley hypothesis.

## 6. Score Matched Actor-Critic (SMAC): regularizing Q-values with dataset scores

SMAC is designed to produce offline checkpoints that online SAC can continue optimizing without first crossing a low-reward region. Its motivation comes from the exact Max-Entropy identity in equation (1): at a SAC optimum, the policy score $\nabla_a \log \pi^*(a|s)$ is proportional to the critic action-gradient $\nabla_a Q^*(s, a)$. We use this identity to identify a structural property that online SAC is naturally driven toward. SMAC then biases the offline critic toward this same structure, so that the resulting checkpoint is better aligned with the optimization path followed during online fine-tuning. We do not assume that the dataset policy is optimal, the learned critic is the dataset policy's critic, or that this identity holds exactly during offline training.

Concretely, SMAC regularizes $\nabla_a Q(s, a)$ toward an estimate of the dataset action score $\nabla_a \log \pi^{\mathcal{D}}(a|s)$. The dataset score is useful because it can be estimated from offline data, and because it provides a local signal for which actions are likely under the data distribution. In this sense, the score estimate is an offline-computable proxy for placing the critic, and the policy induced by its actor loss, in a region more compatible with online actor-critic optimization. This reduces the mismatch between offline pessimistic training and online SAC fine-tuning: instead of placing the offline checkpoint at a high-reward solution separated from SAC's

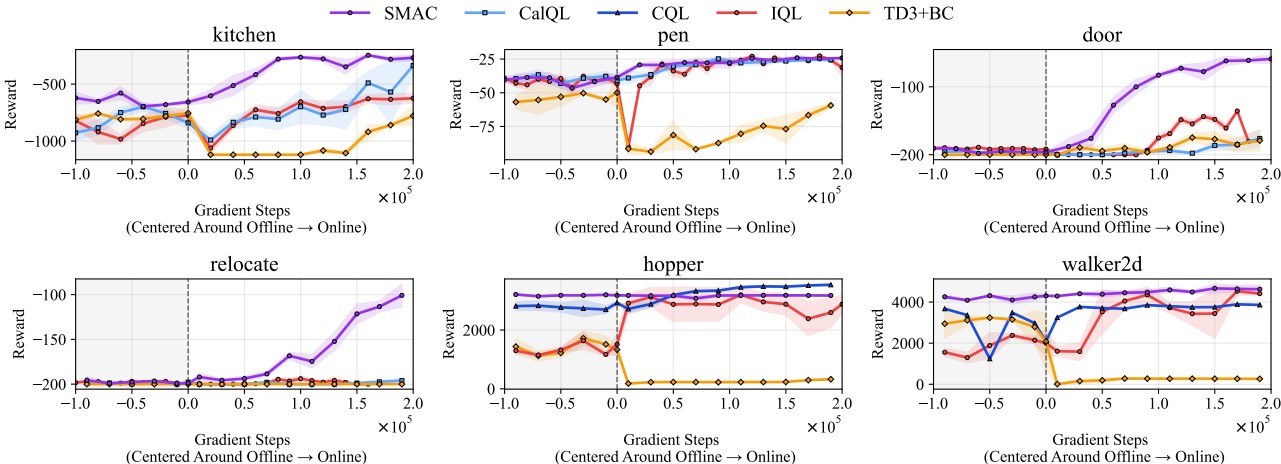

*Figure 5.* **SMAC achieves smooth offline-to-online transfer to SAC**. Plot shows offline-to-online transfer results for SMAC and baselines when fine-tuned with SAC. Environments are only named with the first word since that uniquely identifies them among the six. The dotted line shows where the agent transfers from offline learning to online learning. The shaded region shows the final offline training window before transfer; the first 200k offline gradient steps are omitted from the plot.

trajectory by a low-reward valley, SMAC aims to place it on or near a trajectory that SAC can continue improving smoothly.

Mechanistically, SMAC changes the local geometry of the critic, not just the scale of its Q-values. On actions supported by the offline data, the score-matching term trains $\nabla_a Q(s, a)$ to align with the score-gradient structure that characterizes SAC optima. The actor therefore begins online fine-tuning with a critic whose local action preferences already resemble those produced by online SAC, rather than one that must first be reshaped before improvement can resume. Away from the data, the same gradient field acts as structured pessimism: values slope back toward supported actions instead of being suppressed by a uniform penalty. This gives a concrete reason to expect SMAC checkpoints to lie on or near optimization paths that SAC can continue smoothly, avoiding the low-reward valleys observed for prior offline checkpoints.

The remainder of this section defines how we estimate the dataset score, how the score-matching loss modifies the SAC critic objective, and why we use Muon as an empirical stabilizer during offline training.

### 6.1. Estimating the dataset's score

We employ Reinforcement via Supervision (RvS) methods (Emmons et al., 2022; Piche et al., 2022; Schmidhuber, 2020) for obtaining a strong diffusion policy which in turn provides strong score estimates. Specifically, we condition the estimator to model $\nabla_a \log(p(a|s, w))$ where $w$ is the sum of rewards or binary success indicator of the trajectory that $(s, a)$ belongs to in the dataset. We min-max normalize

$w$ in every dataset so $w = 1$ implies an action in a trajectory with an optimal outcome. For parameterization of our diffusion model, we use the architecture proposed in (Hansen-Estruch et al., 2023). RvS conditioning is important because it biases the score estimate toward high-return actions rather than the average action under the full dataset, which better matches the optimal-policy structure motivating SMAC. Training details are provided in Appendix O.

### 6.2. Regularizing the $Q$-function with score matching

To regularize the critic's network to be proportional to the dataset score, we learn $\alpha$ through a network $\alpha_\psi(s)$ conditioned on states. Conditioning on states allows a more expressive class of networks to minimize the regularization term. Letting $\epsilon_\omega$ be the learned diffusion model we use as a score-estimate, our regularization loss, which we denote $\mathcal{L}^{SM}(\theta, \psi)$, is defined as:

$$\mathcal{L}^{SM}(\theta, \psi) = \mathop{\mathbb{E}}_{\substack{s \sim \mathcal{D} \\ a \sim B(\mathcal{A})}} [|||\nabla_a Q_\theta(s, a) - \alpha_\psi(s)\epsilon_\omega(s, a, w, 1)||_2^2]$$

We set $k = 1$ on our diffusion model to get the least perturbed noise estimate. We define the distribution we sample actions from, $B(\mathcal{A})$, as being a 50/50 split between sampling from $\pi$ and from the uniform distribution over the action space $\mathcal{A}$. $\mathcal{L}^{SM}$ is the only change we make to the original SAC (Haarnoja et al., 2018) objective functions. We follow their implementation by using target Q-networks (Mnih et al., 2013; Lillicrap et al., 2019) and ensembles of Q-functions (Chen et al., 2021; Peer et al., 2021) which are both well-established techniques for reducing Q-function

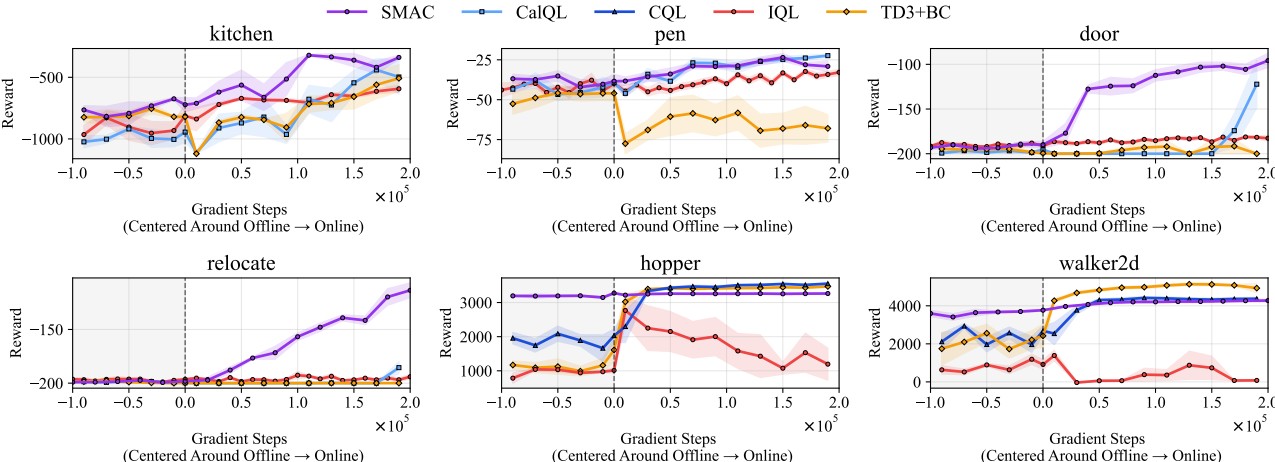

*Figure 6.* **SMAC achieves smooth offline-to-online transfer to TD3**. Plot shows offline-to-online transfer results for SMAC and baselines when fine-tuned with TD3. The format follows Figure 5, including the omission of the first 200k offline gradient steps.

misestimation. For simplicity of notation, we omit the ensembles, but denote the target network as $Q_{\bar{\theta}}$. Using this notation, the original SAC critic loss is:

$$\mathcal{L}^{AC}(\theta) = \mathbb{E}_{\substack{s,a,r,s' \sim D \\ a' \sim \pi_\phi(a|s)}}[(Q_\theta(s,a) - r - \gamma Q_{\bar{\theta}}(s',a'))^2]$$

We augment this loss by adding $\mathcal{L}^{SM}(\theta,\psi)$ multiplied by a coefficient $\kappa$. With this, we define SMAC's critic loss, $\mathcal{L}^{SMAC}$, as:

$$\mathcal{L}^{SMAC}(\theta,\psi) = \kappa \mathcal{L}^{SM}(\theta,\psi) + \mathcal{L}^{AC}(\theta)$$

We optimize the $\pi_\phi$ in SMAC by using the SAC policy loss:

$$\mathcal{L}^\pi(\phi) = \mathbb{E}_{\substack{s \sim D \\ a \sim \pi_\phi}}[-Q_\theta(s,a) + \log \pi_\phi(a|s)]$$

### 6.3. Using Muon as an optimizer

We use Muon (Jordan, 2024; Bernstein & Newhouse, 2024) rather than Adam (Kingma & Ba, 2017) during the offline phase as an empirical stabilizer for the actor-critic checkpoint. Bernstein & Newhouse (2024) show that Adam takes a step in the direction of steepest descent under the max-of-max norm, which is effectively the maximum absolute value of any single parameter. Muon instead takes a step in the direction of steepest descent under the spectral norm, the largest singular value in a matrix. Recent work finds that Muon optimizes toward shallower optima (Anonymous, 2025), a property linked to stronger transfer to downstream fine-tuning (Liu et al., 2023). This makes Muon a plausible optimizer for improving the stability of the offline

checkpoint, while the score-matching regularizer remains the main mechanism that aligns the critic with online actor-critic updates. Appendix N reports the optimizer ablations.

## 7. Experimental Results

### 7.1. Does SMAC remove the initial online performance drop?

Figure 5 shows offline-to-online transfer when fine-tuning with SAC. Most baselines experience drastic performance drops upon transfer: CalQL in 3/4 environments, IQL in 4/6, and TD3+BC in 5/6. In contrast, SMAC avoids any performance drop across all environments, smoothly improving to the highest observed performance.

*Table 1.* Normalized Regret averaged over all six environments ($\downarrow$ lower is better). Values are min-max normalized per environment then averaged. SMAC achieves the lowest regret across all four online algorithms, with particularly strong gains when paired with SAC.

| | Online Algorithm | | | |
|---|---|---|---|---|
| Offline Algorithm | AWR | SAC | TD3 | TD3+BC |
| IQL | 0.508 | 0.471 | 0.653 | 0.494 |
| SMAC | 0.380 | 0.031 | 0.090 | 0.226 |
| TD3+BC | 0.654 | 0.962 | 0.545 | 0.562 |
| CalQL/CQL | 0.482 | 0.448 | 0.442 | 0.614 |

Table 1 reports normalized online regret averaged across the six environments. For each environment, we compute cumulative regret over the evaluated online fine-tuning horizon as $\sum_{t=1}^{T}(R^* - R_t)$, where $R^*$ is the highest reward observed in that environment across all agents and $R_t$ is the reward at online step $t$. We then min-max normalize regret within each environment, with 0 denoting the lowest

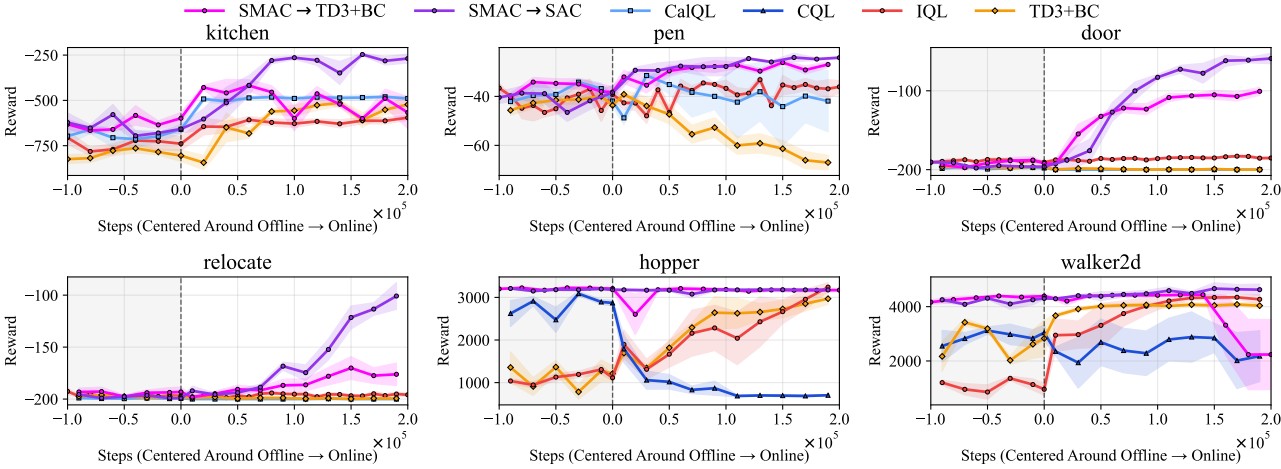

*Figure 7.* **Fine-tuning with TD3+BC stabilizes offline-to-online**. Plot shows offline-to-online transfer results for SMAC and baselines when fine-tuned with TD3+BC. The format follows Figure 5, including the omission of the first 200k offline gradient steps.

observed regret and 1 the highest, before averaging across environments. SMAC achieves the lowest average regret for every online fine-tuning algorithm in Table 1. Across all offline-online combinations, SMAC has the lowest regret in 4/6 environments; in those environments, its regret is 34% to 58% lower than the strongest baseline. In the remaining two environments, SMAC attains the second- and third-lowest regret. Per-environment regret values are reported in Appendix G.

### 7.2. Does connectivity coincide with offline-to-online performance?

These transfer results correlate well with the interpolation and planar plots from Section 5. In the tasks where we evaluate both connectivity and transfer, the presence of a low-reward valley between offline and SAC-fine-tuned checkpoints coincides with an initial drop in online fine-tuning. Conversely, methods with monotone interpolation curves transfer without the same initial collapse. This correspondence supports our use of connectivity as a diagnostic for offline-to-online stability, while leaving the full connectivity analysis in Section 5.

### 7.3. Does SMAC transfer across online fine-tuning algorithms?

We compare different offline RL algorithms' generated offline checkpoints' ability to transfer to online RL algorithms by transferring to: SAC, TD3, and TD3+BC. SAC is the most representative and popular algorithm in online value-based RL combining deterministic gradients with entropy regularization. TD3 is an antecedent algorithm to SAC which doesn't use entropy regularization and only uses deterministic gradients for updating the policy. For this

reason we include it, Lee et al. (2025) similarly evaluate their architecture against SAC and DDPG (antecedent algorithm to TD3 which only uses deterministic gradients) to show the architecture works well along popular online value-based techniques. TD3+BC adds behaviour cloning to TD3, while originally an offline RL algorithm and less efficient than TD3, it has been shown to improve transfer abilities for offline-to-online RL (Dong et al., 2025b). We also evaluate Advantage Weighted Regression (AWR), a behavior-weighted policy improvement method that updates the actor toward replay-buffer actions weighted by estimated advantage. Since AWR is closer to offline policy optimization than standard online actor-critic fine-tuning, we treat it as supporting evidence and report the full discussion in Appendix H. We don't consider online transfer to SMAC because the pre-trained diffusion model would then have to be continually updated which would be both computationally costly and can result in catastrophic forgetting in important states and actions.

In Figure 6 we plot all methods fine-tuned with TD3. Similarly, Figure 7 plots all methods fine-tuned with TD3+BC, with SMAC→SAC overlaid as a lower bound for optimal adaptive performance. SMAC still achieves smooth transfer in 6/6 environments when fine-tuning with TD3 and 4/6 when fine-tuning with TD3+BC. However in 2 environments (kitchen and walker2d) SMAC's performance deteriorates with ongoing training. We attribute this to the BC term incentivizing the policy to copy suboptimal actions in the dataset and replay buffer. This is supported by the fact that pen, door, and relocate (which contain only successful demonstrations) show no such degradation.

From comparing Figures 6 and 7 we observe that in some environments, adding a BC term stabilizes offline-to-online transfer for IQL and TD3+BC while causing long term per-

formance drops in SMAC and CalQL/CQL not observed when using TD3. One can also separate IQL and TD3+BC from CalQL/CQL and SMAC by whether they constrain the policy during the offline phase. During the online phase, regularizing the actor-critics generated by IQL and TD3+BC close to the behaviour data is a continuation of their offline optimization; this is the opposite for CalQL/CQL and SMAC. Conversely, fine-tuning with TD3 where the agents must only maximize the critic's estimate of the policy's sampled actions closely resembles the offline policy optimization for CalQL/CQL and SMAC. We believe this dichotomy between fine-tuning with TD3 vs. TD3+BC and which offline algorithms pair better reinforces our hypothesis about connected optima being an explanatory factor for offline-to-online stability. Appendix I further isolates this compatibility effect in Kitchen by comparing IQL → IQL and CalQL → CalQL against SAC fine-tuning from the same offline checkpoint families.

### 7.4. How do sharpness and batch size carry over into offline-to-online RL?

We study batch-size effects in Appendix D. These experiments ask whether sharpness-motivated optimization effects, which are often studied through batch size and learning-rate ratios, also affect offline-to-online transfer. We find that larger offline batches improve SMAC's offline performance, while larger online batches improve adaptation speed; however, batch size alone does not remove the baseline transfer failures.

### 7.5. Which implementation choices matter?

We ablate two implementation choices in the appendix: RvS conditioning for the diffusion score estimator and the Muon optimizer. Appendix E shows that removing RvS weakens transfer in several environments, and Appendix N shows that replacing Muon with Adam weakens SMAC while adding Muon to the baselines does not reproduce SMAC's transfer stability. These ablations suggest that the score-matching regularizer is the main mechanism, while RvS and Muon are important stabilizers.

## 8. Related works

SMAC follows previous work showing that strong offline RL agents can be trained by penalizing the Q-function on OOD actions (Kumar et al., 2020; Jin et al., 2022; Wen et al., 2024; Kostrikov et al., 2021; Yu & Zhang, 2023; Zhou et al., 2024; Nakamoto et al., 2023). This regularization ensures that the critic disincentivizes the policy from choosing OOD actions. A different branch of algorithms ensures this by regularizing the policy to stay close to dataset actions (Fujimoto & Gu, 2021; Wu et al., 2019; Kostrikov et al., 2022; Nair et al., 2020). As discussed in Section 5 the first type of

approach mis-specifies the problem while the second under-specifies it. In our results section, we show that algorithms which follow the first branch exhibit better offline-to-online performance, implying that mis-specification is better than under-specification for offline-to-online transfer.

Unlike our method, several offline-to-online papers have looked at algorithms which are applied in both the offline and online phases. These methods rely on regularizing the critic (Nakamoto et al., 2023; Lee et al., 2021; Yu & Zhang, 2023; Zheng et al., 2023), policy (Zhang et al., 2023a;b), or both (Wen et al., 2024). We differentiate our method from this collection of work as our focus is purely on offline RL algorithms which transfer to general actor-critic algorithms like SAC or TD3. This focus is shared with Yu & Zhang (2023) and Zhao et al. (2023). Zhao et al. (2023) advocate CQL with large critic ensembles for the offline stage. Since our CalQL/CQL baseline already uses an ensemble critic, we expect it to capture much of the benefit of their proposal.

Kostrikov et al. (2021) and Yu & Zhang (2023) are the methods most similar to SMAC. Both papers leverage the exact Max-Entropy RL identity to design offline RL algorithms whose Q-value estimates are influenced by the score of the dataset. Our work is most similar to Yu & Zhang (2023) who first learn a Q-function that is parameterized by a value function plus the score of the policy, similar to Gu et al. (2016). Yu & Zhang (2023) extends beyond Gu et al. (2016) because at the start of online training, they extend the width of the first layer of the value network to incorporate actions as well. We chose not to include Yu & Zhang (2023) as a baseline because, while conceptually similar, their parameterization means that the offline algorithm does not return an actor network and a critic network which can be used straightforwardly by SAC or TD3+BC. Additionally, their experimental results show unstable transfer and suboptimal regret as compared to CQL fine-tuned with SAC.

## 9. Limitations & future work

SMAC fits naturally with the current direction of robot learning, where large behavior-cloned generative action models and VLAs are increasingly used as pretrained robot policies (Black et al., 2024; Chi et al., 2024; Intelligence et al., 2025; NVIDIA et al., 2025; Team et al., 2025; Shukor et al., 2025). These models already learn rich action distributions from broad robot datasets, so they provide a natural substrate for estimating the score signals SMAC needs. A promising direction is to reuse or fine-tune pretrained action models, allowing SMAC-style actor-critic pretraining to build on existing generative policies rather than training a score model from scratch for every task.

SMAC is motivated most directly by the structure of entropy-regularized actor-critic methods such as SAC. Our experi-

ments show that the resulting checkpoints transfer well to SAC, TD3, TD3+BC, and AWR, but the method does not yet guarantee smooth transfer to arbitrary online RL algorithms. A useful direction for future work is to characterize which online update rules preserve the same score-gradient geometry, and which require different offline regularizers.

SMAC also pays an upfront cost to estimate the dataset score. In our implementation, training the diffusion score model adds roughly 3–4 GPU hours on a single L40S, and the full SMAC offline phase takes about 3x the wall-clock time of IQL and 2x that of CalQL. Appendix K shows that giving the strongest baselines comparable extra offline compute does not close the Kitchen transfer gap, but reducing SMAC's score-estimation cost remains important. Promising directions include reusing pretrained diffusion policies, estimating scores with lighter models, or directly regressing the critic's action-gradient without a separate diffusion model.

Finally, our connectivity evidence is strongest for the benchmark families studied here. The additional Door, Hopper, and Cube experiments broaden the empirical picture, but connectedness of offline and online optima should be tested across more environment structures, datasets, and online update rules. Smooth transfer also remains sensitive to online batch size, as shown in Appendix D. This suggests that future offline-to-online methods should treat checkpoint geometry and online optimization choices as coupled design problems.

## 10. Conclusion

Offline RL can pre-train strong actor-critics, but those checkpoints are only useful for deployment if online fine-tuning can improve them without first causing a performance collapse. This paper argues that the drop is geometric: prior offline methods often find high-reward checkpoints that are separated from online actor-critic optima by low-reward regions, so fine-tuning must cross a bad part of parameter space before recovering. SMAC addresses this by shaping the offline critic toward the score-gradient structure satisfied by online SAC optima, producing checkpoints that are more compatible with later actor-critic updates. Across 6 D4RL tasks, SMAC transfers smoothly to SAC and TD3, and in 4/6 environments reduces online regret by 34–58% relative to the best baseline.

## Impact Statement

This paper presents work whose goal is to advance the field of machine learning. There are many potential societal consequences of our work, none of which we feel must be specifically highlighted here.

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

## A. Reinforcement Learning

We assume the traditional Markov Decision Process (MDP) formulation of RL, which optimizes policies $\pi$ towards maximizing the future discounted sum of rewards. An MDP, commonly referred to as an environment, is defined as a 6-tuple $(\mathcal{S}, \mathcal{A}, R_{s,a}, T_{s,a}^{s'}, d_0, \gamma)$ where: $\mathcal{S}$ is a state space, $\mathcal{A}$ an action space, $R_{s,a} : \mathcal{S} \times \mathcal{A} \to \mathbb{R}$ a reward function, $T_{s,a}^{s'} : \mathcal{S} \times \mathcal{A} \to \Delta(\mathcal{S})$ a transition function which takes a state-action pair and returns a next state, and $d_0$ an initial state distribution. A policy $\pi : \mathcal{S} \to \Delta(A)$ maps states to distributions over actions. The optimization objective for RL is to find $\pi$ which maximizes $\mathcal{J}(\pi) = \mathbb{E}_{s_0 \sim d_0}[\sum_{t=1}^{\infty} \gamma^t R(s_t, a_t) | s_t \sim T_{s,a}^{s'}(s_t|s_{t-1}, a_{t-1}), a_t \sim \pi(a_t|s_t)]$ which is the expected return from following $\pi$ in the MDP. For notation, we use $s$ for states, $a$ for actions, $r$ for rewards, $\pi(a|s)$ for policies, $G_t$ for the discounted sum of rewards following time-step $t$, $\mathbb{E}_\pi[\sum_t^{\infty} \gamma^t R(s_t, a_t)]$ as the expected sum of rewards following $\pi$. We denote the policy that generates a dataset $\mathcal{D}$ by $\pi^D$ and define datasets as a set of state, action, reward, and next state tuples: $\mathcal{D} = \{(s_i, a_i, r_i, s_i')\}_{i=1}^N$.

## B. Value-based Reinforcement Learning

Value-based RL is an efficient approach for learning optimal policies from on-policy and off-policy experience. It involves learning a Q-function $Q^\pi(s, a)$, also called a critic, for a policy $\pi$ which takes the current state $s$ and action $a$ and attempts to predict the discounted sum of future rewards $\mathbb{E}_\pi[\sum_{t=0}^T \gamma^t R(s_t, a_t) | a_0 = a, s_0 = s]$. We craft prediction targets for learning a Q-function through the temporal difference update that leverages the following identity:

$$\mathbb{E}_\pi[\sum_{l=t}^{\infty} \gamma^l r_l] = r_t + \gamma \mathbb{E}_\pi[\sum_{l=t+1}^{\infty} \gamma^{l-t} R(s_l, a_l)]$$
$$= r_t + \gamma \mathbb{E}_{\substack{s' \sim T_{s,a}^{s'}(s,a) \\ a' \sim \pi(a'|s)}}[Q(s', a')]$$

The Q-function is then learned by minimizing $\mathbb{E}_\pi[(Q(s, a) - r(s, a) - \gamma Q(s', a'))^2]$. Value-based RL uses the learned Q-function to update the policy. This approach relies on Q-function being accurate in the states and actions where the policy queries it during learning. Hence, inaccurate critics can cause divergences in learning because the policy can be optimized towards sampling bad actions that the critic has mis-estimated.

## C. Offline Reinforcement Learning

Offline RL assumes access to only a dataset $\mathcal{D} = \{(s_i, a_i, r_i, s_i')\}_{i=1}^N$ and prohibits the policy from directly interacting with the MDP. Value estimation errors are the main problem that offline RL methods attempt to overcome. In the temporal difference update, the policy is able to pick unseen actions for $a'$. This causes the created prediction target to depend on the Q-function's out-of-distribution predictions. If these out-of-distribution predictions are over-estimates, a biased and incorrectly high target will be formed for our Q-value. This can, in turn, reinforce the policy to sample bad out-of-distribution actions because the Q-function over-estimates their value. To avoid this, the majority of offline RL methods either train the policy to generate actions as close to the data as possible (Fujimoto & Gu, 2021; Kostrikov et al., 2022), restrict the policy from sampling actions in creating the TD-target, or use *pessimism*, which explicitly minimizes out-of-distribution Q-value predictions (Kumar et al., 2020; Nakamoto et al., 2023).

## D. Batch size and notions of sharpness matter in offline-to-online RL

Optimization landscapes have shown that the ratio between batch size and learning rate impacts the generalization strength of the final parameters (He et al., 2019; Elgharabawy et al., 2020). The best generalizing offline RL actor-critic should generalize to the online setting without suffering from distribution shift. So, we investigate whether the findings transfer over to the offline-to-online setting. We vary the ratio by varying the batch size and observe how it relates to the performance drops in offline-to-online RL.

In Figure 8, we ablate over several offline batch sizes while keeping an online batch size of 512 constant. We observe that larger batch sizes uniformly lead to better offline performance. Interestingly, we observe that across methods, the performance drops to a similar point. For the baselines, this common point is worse than any offline performance value.

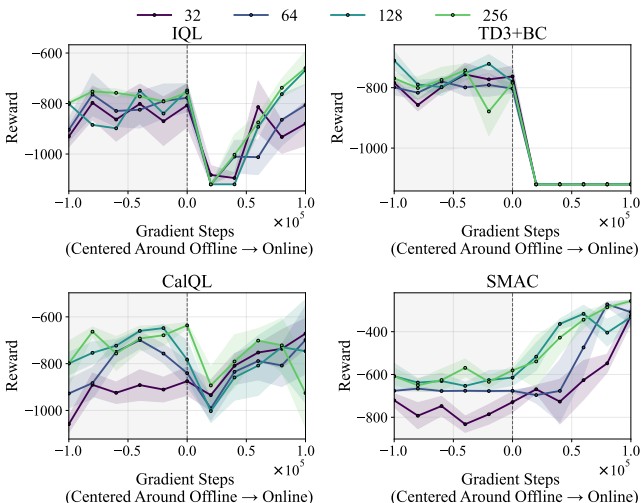

*Figure 8.* **Offline Batch Size has negligible effect on failing baselines, but impacts SMAC**

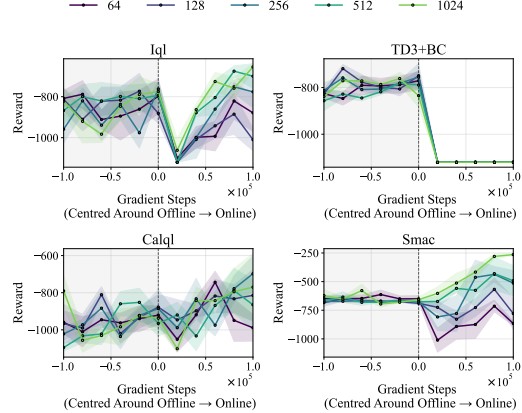

*Figure 9.* **Larger batch sizes lead to faster online adaptation**

However, for our method, that point is the worst offline performance value we observe over batch sizes, and it is often higher than or equal to the baselines' observed offline performances.

Following this observation about minimum batch size, we set an offline batch size of 32 and ablate the online batch size across methods. We plot the results in Figure 9. Similarly to Zhou et al. (2024), we observe that generally larger batch sizes lead to stronger performance, except for the case of TD3+BC, where we continue to see that all batch size choices lead to a drop in performance which is not recovered from within the number of online steps we allotted for the experiment.

## E. Ablating Reinforcement via Supervision in Offline-to-Online performance

In the figure below we plot the performance of SMAC when we use RvS to train and inference the diffusion model vs. when we train the diffusion model without any extra conditioning. We observe that the performance is still significantly stronger than the baselines but suffers slight drops in performance when transferring to online learning across the environments. We also observe that the offline performance is worsened as well. Given that the three environments are ones where the main assumption underlying SMAC is broken, this puts more evidence behind the claim that RvS is a key component allowing SMAC to work when datasets fail the key assumption that $\nabla_a \log \pi(a|s) \propto \nabla_a Q(s, a)$.

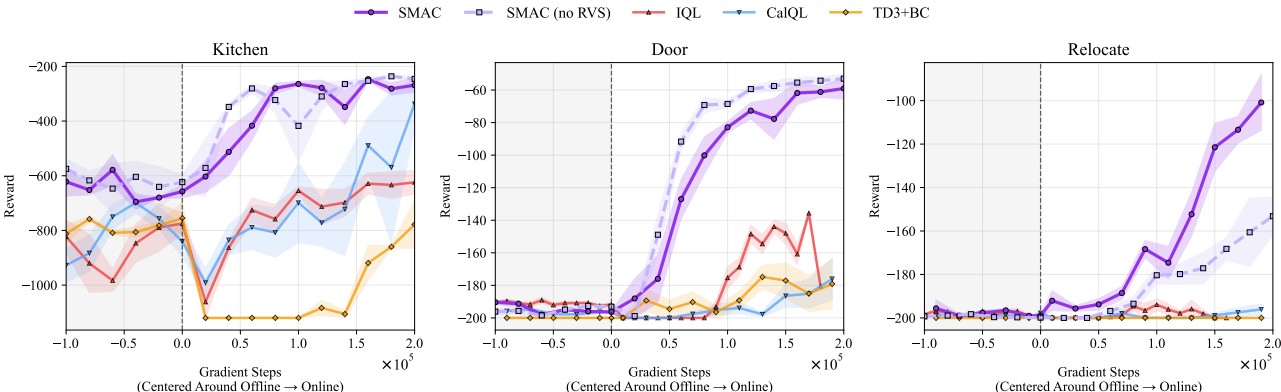

*Figure 10.* Removing RvS hurts transfer in 2/3 environments but has mixed impact on online performance. SMAC still dominates baselines without using RvS for training the diffusion model

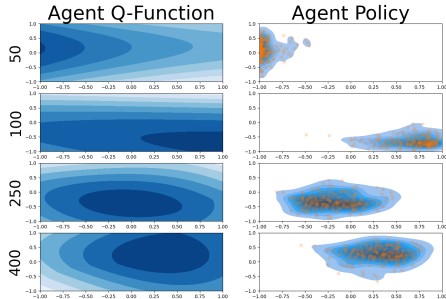

*Figure 11.* **Q-function and score of policy are proportional throughout training**. We plot the Q-function and policy for a SAC agent after 50, 100, 250, and 400 episodes of training in the Reacher-v2 environment.

# F. Verifying Score-Matching Identity Throughout Training

In Figure 11 we plot $\pi(a|s)$ and $Q(s, a)$ in an environment where $\mathcal{A}$ is 2-dimensional. We plot $\pi(a|s)$ and $Q(s, a)$ at several checkpoints during training. The figure shows that identity (1) appears to roughly hold during online training.

# G. Regret during online phase

For each environment we find the max reward observed over all runs and denote it $R^*$. For a run with observed online rewards $\{R_t\}_{t=1}^{\infty}$ we define its regret at $\frac{1}{T}\sum_{t=1}^{T} R^* - R_t$. In the tables below we list the regret of each offline-online algorithm pairing per environment. We write the mean and standard error of the regret. We also shade cells gold, silver, and bronze to signify lowest regret, second lowest regret, and third lowest regret per environment.

*Table 2.* Offline-to-online regret when fine-tuning with SAC ($\downarrow$ lower is better). Entries are mean $\pm$ std over seeds; "-" indicates not evaluated. Cell colors are assigned *per environment across all four tables*: best (gold), second (silver), third (bronze).

| Environment | Offline Algorithm | | | |
| --- | --- | --- | --- | --- |
| | CalQL/CQL | IQL | SMAC | TD3+BC |
| door | $134.5 \pm 1.6$ | $120.0 \pm 0.8$ | $50.3 \pm 2.7$ | $129.7 \pm 2.4$ |
| hopper | $293.7 \pm 20.4$ | $798.0 \pm 126.5$ | $386.3 \pm 10.8$ | $3213.3 \pm 16.3$ |
| kitchen | $467.1 \pm 38.1$ | $492.9 \pm 13.7$ | $131.4 \pm 13.0$ | $762.5 \pm 11.4$ |
| pen | $8.0 \pm 0.7$ | $10.3 \pm 0.7$ | $5.3 \pm 0.7$ | $55.2 \pm 1.9$ |
| relocate | $98.1 \pm 0.4$ | $97.4 \pm 0.4$ | $62.8 \pm 2.1$ | $99.2 \pm 0.0$ |
| walker2d | $1553.7 \pm 0.0$ | $1801.2 \pm 180.6$ | $650.5 \pm 39.9$ | $4739.6 \pm 125.8$ |

*Table 3.* Offline-to-online regret when fine-tuning with TD3 (↓ lower is better). Entries are mean ± std over seeds; "-" indicates not evaluated. Cell colors are assigned *per environment across all four tables*: best (gold), second (silver), third (bronze).

| Environment | Offline Algorithm | | | |
|---|---|---|---|---|
| | CalQL/CQL | IQL | SMAC | TD3+BC |
| door | 131.1 ± 3.0 | 126.1 ± 0.4 | 65.5 ± 2.1 | 137.7 ± 1.4 |
| hopper | 311.4 ± 46.5 | 1834.3 ± 150.3 | 297.3 ± 6.5 | 329.3 ± 39.0 |
| kitchen | 526.6 ± 32.3 | 445.9 ± 13.2 | 259.3 ± 22.8 | 529.1 ± 21.6 |
| pen | 8.4 ± 0.7 | 16.1 ± 0.5 | 8.5 ± 0.8 | 41.6 ± 2.9 |
| relocate | 97.8 ± 0.5 | 95.2 ± 0.3 | 58.3 ± 1.3 | 99.1 ± 0.0 |
| walker2d | 1148.2 ± 63.4 | 4689.0 ± 101.3 | 987.5 ± 17.2 | 454.9 ± 62.2 |

*Table 4.* Offline-to-online regret when fine-tuning with AWR (↓ lower is better). Entries are mean ± std over seeds; "-" indicates not evaluated. Cell colors are assigned *per environment across all four tables*: best (gold), second (silver), third (bronze).

| Environment | Offline Algorithm | | | |
|---|---|---|---|---|
| | CalQL/CQL | IQL | SMAC | TD3+BC |
| door | 129.9 ± 0.7 | 127.9 ± 0.4 | 122.8 ± 0.8 | 136.0 ± 0.7 |
| hopper | 533.7 ± 43.0 | 958.0 ± 89.9 | 353.0 ± 2.6 | 552.4 ± 26.8 |
| kitchen | 343.7 ± 7.3 | 446.4 ± 10.0 | 340.6 ± 15.8 | 837.0 ± 3.7 |
| pen | 28.8 ± 2.6 | 18.0 ± 0.5 | 13.3 ± 0.7 | 32.9 ± 3.0 |
| relocate | 99.0 ± 0.1 | 95.8 ± 0.3 | 98.3 ± 0.2 | 98.1 ± 0.2 |
| walker2d | 1136.9 ± 42.6 | 1918.7 ± 114.4 | 544.4 ± 58.1 | 1988.4 ± 312.6 |

# H. Fine-tuning with AWR

Advantage Weighted Regression (AWR) is a policy improvement method that updates the actor by fitting actions from a replay buffer, weighted by their estimated advantages. Unlike SAC or TD3, AWR does not directly optimize sampled policy actions against the critic; instead, it performs a behavior-weighted update that keeps the policy closer to actions observed in the data. This makes AWR a natural stress test for our compatibility hypothesis: it can reduce abrupt transfer drops by continuing a behavior-regularized objective, but it may also limit the speed and final quality of online improvement.

Figure 12 shows that this tradeoff appears in our experiments. AWR fine-tuning stabilizes several baselines, but the resulting learning curves are generally lower than those obtained with SAC, TD3, or TD3+BC. SMAC still transfers smoothly under AWR and attains lower average regret than the baselines, but AWR fine-tuning is generally less efficient than the main online fine-tuning algorithms evaluated in the body of the paper.

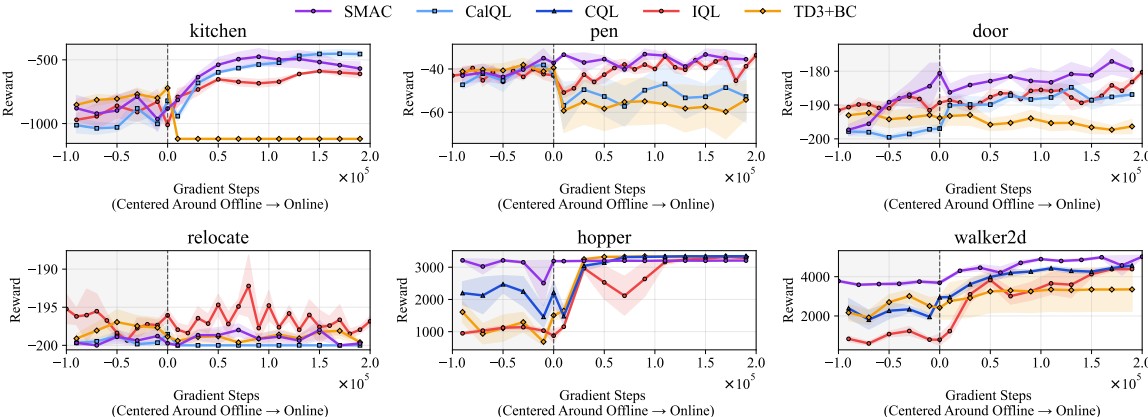

*Figure 12.* **Fine-tuning with AWR is stable but less efficient**. Offline-to-online curves when fine-tuning with AWR. SMAC achieves smooth transfer in 4/6 environments and dominates the baselines in 5/6 environments, but AWR fine-tuning generally produces lower reward curves than the main online fine-tuning algorithms.

*Table 5.* Offline-to-online regret when fine-tuning with TD3+BC (↓ lower is better). Entries are mean ± std over seeds; "-" indicates not evaluated. Cell colors are assigned *per environment across all four tables*: best (gold), second (silver), third (bronze).

| | Offline Algorithm | | | |
| --- | --- | --- | --- | --- |
| Environment | CalQL/CQL | IQL | SMAC | TD3+BC |
| door | $140.8 \pm 0.1$ | $127.0 \pm 0.3$ | $72.3 \pm 1.8$ | $140.3 \pm 0.2$ |
| hopper | $2469.9 \pm 42.3$ | $1392.6 \pm 118.4$ | $425.5 \pm 44.3$ | $1295.4 \pm 62.3$ |
| kitchen | $256.6 \pm 4.0$ | $385.2 \pm 7.6$ | $262.2 \pm 14.6$ | $373.4 \pm 11.5$ |
| pen | $17.8 \pm 2.9$ | $16.1 \pm 0.6$ | $7.7 \pm 0.6$ | $31.8 \pm 1.0$ |
| relocate | $99.1 \pm 0.0$ | $95.3 \pm 0.3$ | $84.9 \pm 2.0$ | $98.2 \pm 0.2$ |
| walker2d | $2642.5 \pm 235.6$ | $1548.6 \pm 118.0$ | $1232.3 \pm 194.5$ | $1242.6 \pm 64.8$ |

## I. Same-objective fine-tuning in Kitchen

A central question in offline-to-online RL is whether transfer instability comes from the offline checkpoint itself, or from a mismatch between the offline objective that produced the checkpoint and the online objective used for fine-tuning. To separate these effects, we compare same-objective fine-tuning in `kitchen-partial-v0` against SAC fine-tuning from the same families of offline checkpoints. Figure 13 shows that same-objective fine-tuning with IQL and CalQL avoids the sharp initial collapse seen when those checkpoints are fine-tuned with SAC, but also improves substantially more slowly than SMAC fine-tuned with SAC. This supports the view that objective compatibility matters for smooth transfer, while also showing why simply continuing the offline objective is not enough for efficient online adaptation.

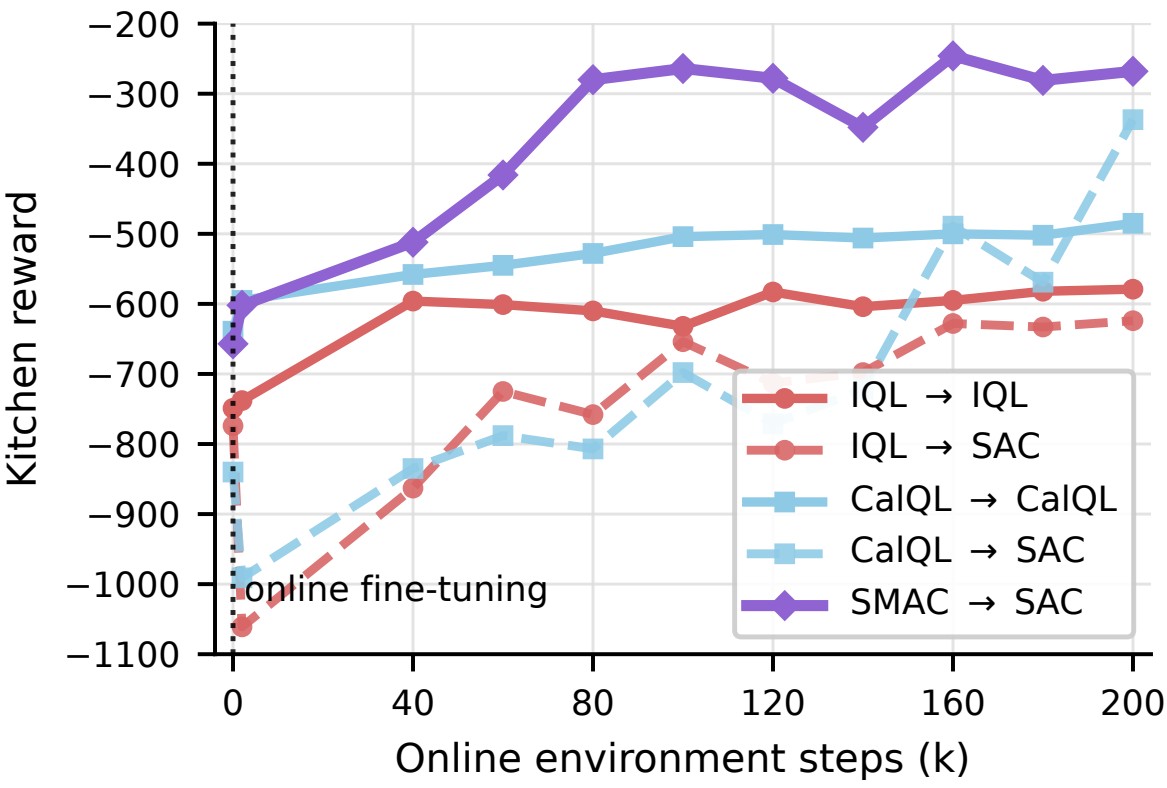

*Figure 13.* **Same-objective fine-tuning is stable but slower in Kitchen**. We compare same-objective fine-tuning for IQL and CalQL against SAC fine-tuning from the same offline checkpoint families. Continuing the offline objective avoids the sharp initial SAC drop, but SMAC fine-tuned with SAC improves more quickly and reaches stronger final performance.

We also evaluate linear interpolation between the offline checkpoint and the checkpoint obtained after same-objective fine-tuning. Figure 14 shows that the IQL → IQL and CalQL → CalQL endpoints are linearly connected in Kitchen: reward

does not pass through the kind of low-reward valley observed when these offline checkpoints are fine-tuned with SAC. Together with Figure 13, this supports the interpretation that same-objective fine-tuning is stable because the offline and online endpoints are better aligned, but that this stability comes at the cost of slower online improvement.

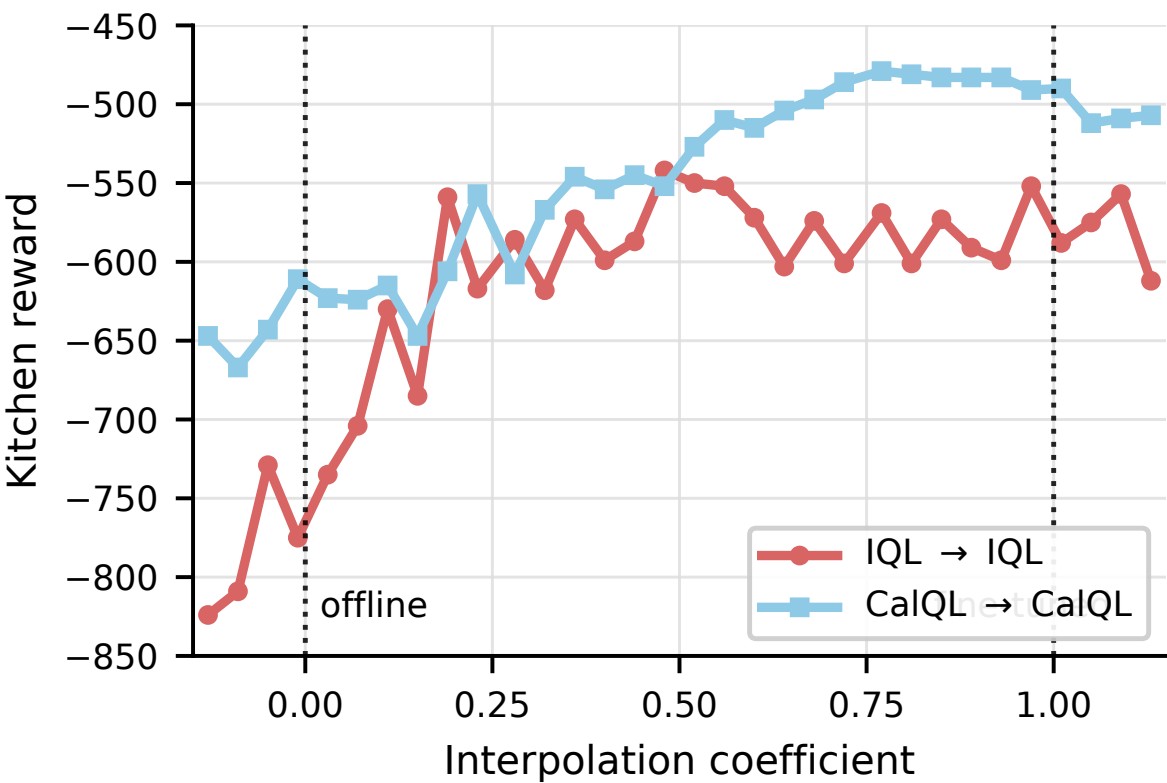

*Figure 14.* **Same-objective Kitchen fine-tuning produces linearly connected endpoints**. We plot reward along the line between the offline checkpoint and the same-objective fine-tuned checkpoint. Both IQL → IQL and CalQL → CalQL avoid a low-reward interpolation valley, matching their stable online learning curves in Figure 13.

## J. Transfer with a large Door dataset

A high offline reward does not by itself guarantee smooth online transfer. To separate transfer stability from offline dataset coverage, we evaluate Door with a dataset containing over 10 million samples. This dataset is large enough that the offline methods obtain strong and relatively similar pre-transfer performance. Figure 15 shows that the baselines can nevertheless suffer sharp drops at the start of online fine-tuning, while SMAC improves immediately. This supports the view that the geometry of the learned actor-critic, not only its offline performance or the amount of data used to train it, determines whether the checkpoint can be fine-tuned smoothly.

## K. Compute-matched Kitchen comparison

SMAC incurs additional offline compute because it trains a diffusion model for score estimation. In our experiments, this diffusion model is trained once on the static dataset and adds roughly 3–4 GPU hours on a single L40S. The full SMAC offline phase takes approximately 3x the wall-clock time of IQL and 2x the wall-clock time of CalQL on the same hardware. To make the comparison more conservative, we train IQL for 3x as many offline gradient steps and CalQL for 2x as many offline gradient steps in Kitchen, then fine-tune the resulting checkpoints with SAC. Figure 16 shows that additional offline compute does not remove the transfer gap: compute-matched IQL and CalQL still suffer from poor online transfer relative to SMAC.

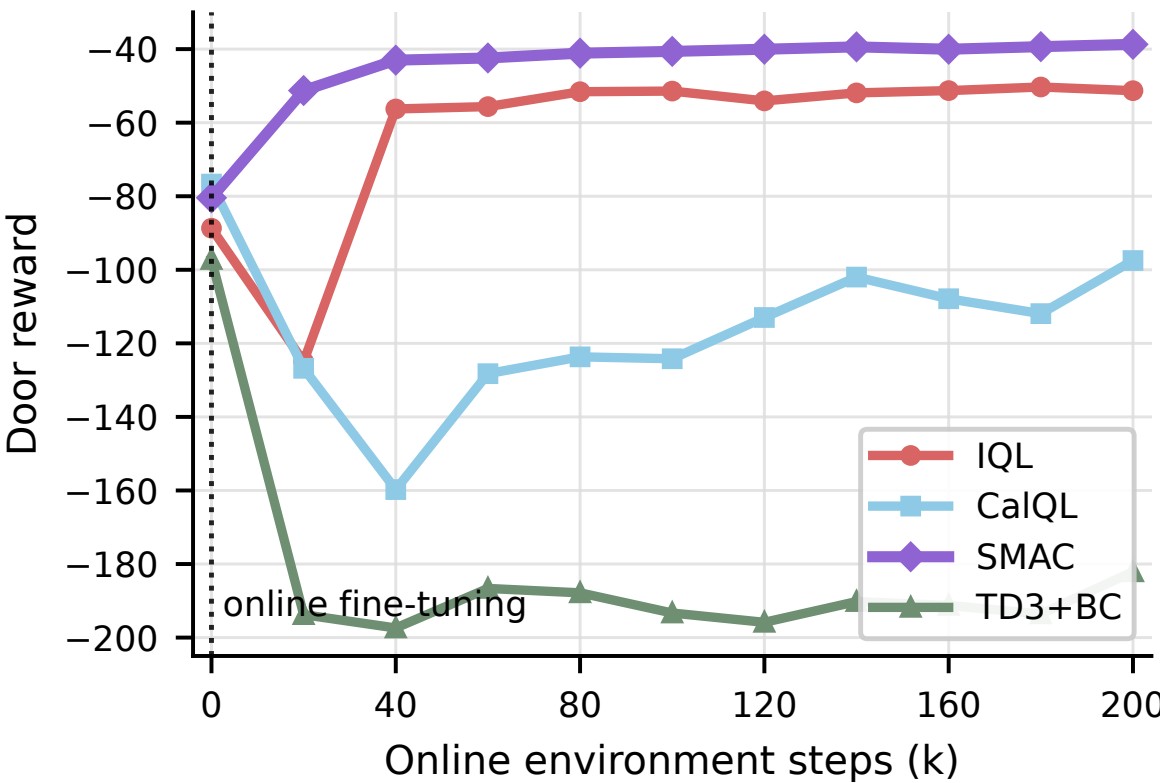

*Figure 15.* **Large high-coverage Door data does not remove transfer instability**. We fine-tune actor-critics trained on a Door dataset with over 10 million samples. Despite strong offline performance, the baselines still experience sharp online drops, while SMAC improves immediately after transfer.

## L. Additional transfer results in Hopper and Cube

We include two additional transfer experiments to test SMAC outside the main benchmark set. Hopper-medium contains suboptimal demonstrations without extreme failure trajectories, so online fine-tuning must improve the policy without first falling into a failure mode. OGBench Cube is qualitatively different: its behavior data is generated by preset motions, many of which are unrelated or counterproductive to the downstream task. This weakens the assumption that the behavior policy is itself well described by an actor-critic optimum. Figure 17 shows that SMAC remains stable in both settings, suggesting that the regularizer is useful even when the dataset only approximately matches the assumptions behind the score-gradient motivation.

## M. Sensitivity to $\kappa$

In Figure 18, we evaluate SMAC in Kitchen across a range of score-matching weights $\kappa$. We selected the final values in Table 7 by a coarse manual sweep, choosing values that reliably produced strong offline performance without an initial online drop. The sweep suggests that SMAC is not sensitive to fine tuning of $\kappa$ once the regularizer is strong enough: performance improves as $\kappa$ increases from very small values, then largely plateaus. The main failure mode is under-regularization. At $\kappa = 5$, SMAC obtains worse offline performance and shows a small drop at online transfer, while for $\kappa < 5$ the offline policy fails to reach high reward.

## N. How much does adding Muon matter?

In Figure 19, we plot the considered baselines' performance when optimized with Muon in the `kitchen-partial-v0` environment. We do this to answer whether Muon was the only key improvement that led to the stable transfer we get from

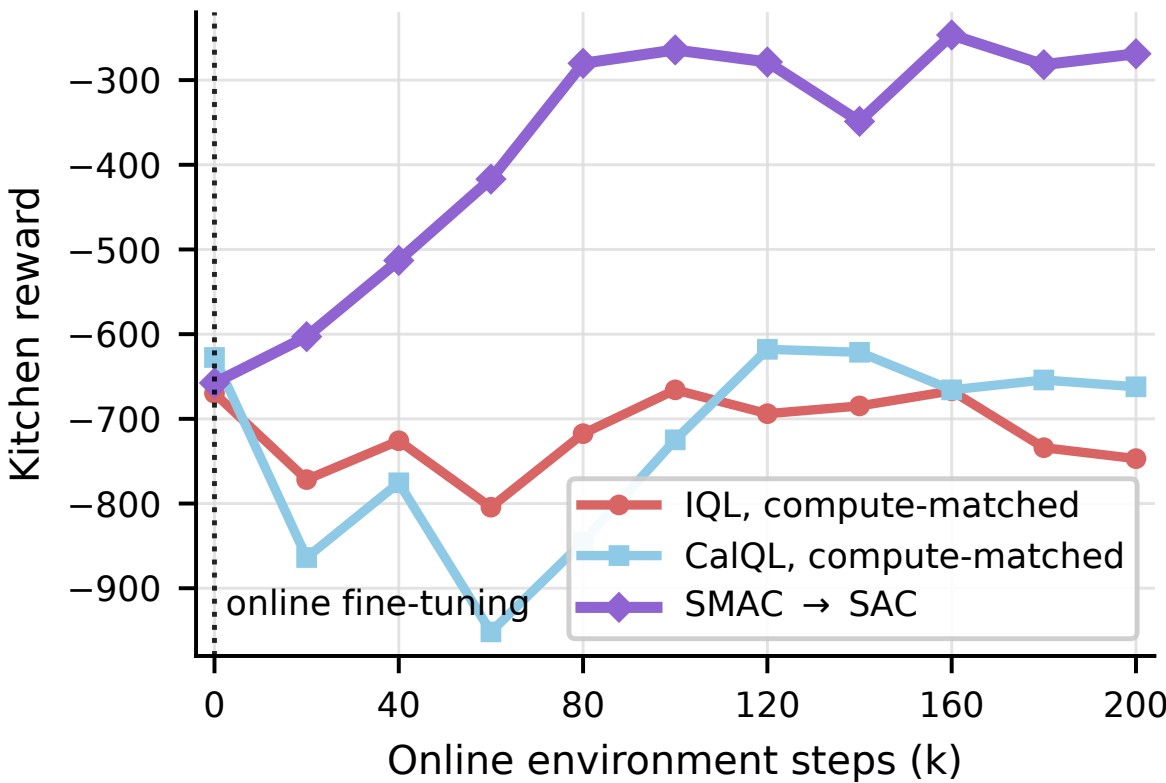

*Figure 16.* **Compute matching does not close the Kitchen transfer gap**. We compare SMAC to IQL and CalQL checkpoints trained with additional offline gradient steps to roughly match SMAC's offline wall-clock cost. The compute-matched baselines still transfer substantially worse than SMAC fine-tuned with SAC.

SMAC.

Optimizing with Muon leads to better offline performance, but worse transfer stability with IQL and TD3+BC experiencing complete policy collapse. Previous papers studying the use of Muon and other "whitening" optimizers show that Muon converges to maxima which exhibit smaller condition numbers in the loss Hessian and take steps that minimize the spectral norm of the gradients (Bernstein & Newhouse, 2024; Frans et al., 2025). This leads the network to converge to flatter maxima, which are in turn harder to escape from if trying to optimize towards a different maximum (Ibayashi & Imaizumi, 2023; Kleinberg et al., 2018).

We now plot the results across the 6 environments when we optimize SMAC with Adam instead of Muon. We find that SMAC still transfers in 3/6 environments but performance suffers across nearly all environments except Hopper.

## O. Diffusion Model Introduction, Training, and Hyper-parameters

A diffusion model $\epsilon_\theta(x)$ learns to match noised versions of $\nabla \log p(x)$ at $K$ varying noise levels (Song & Ermon, 2020; Ho et al., 2020). In practice, a diffusion model is trained to reverse the K-step noising process $x^k = \sqrt{\bar{\alpha}_k} x^{k-1} + \sqrt{1 - \bar{\alpha}_k} \epsilon^k$ where, $\epsilon \sim N(0, I)$ and $\{\alpha_k\}_{k=1}^K$ define the noise schedule. The noise schedule can be any monotonically decreasing function, but, for our experiments we used the cosine noise schedule (Ramesh et al., 2021). The reverse denoising step is $x^{k-1} = (x^k - \sqrt{1 - \bar{\alpha}_k} \epsilon^k)/\sqrt{\bar{\alpha}_k}$. The $\bar{\alpha}_k$ noise schedule is assumed to be known and, so the diffusion model is trained to predict $\epsilon$ given $\sqrt{\bar{\alpha}_k} x + \sqrt{1 - \bar{\alpha}_k} \epsilon$ and $k$ as input. The loss for a diffusion model $\epsilon_\theta$ is

$$\mathbb{E}_{x \sim D, \epsilon \sim I, k \sim U[k]}[||\epsilon - \epsilon_\theta(\sqrt{\bar{\alpha}_k} x + \sqrt{1 - \bar{\alpha}_k} \epsilon, k)||_2^2]$$

This loss function is equivalent to the noised conditional score network loss (Song & Ermon, 2020). So, the converged

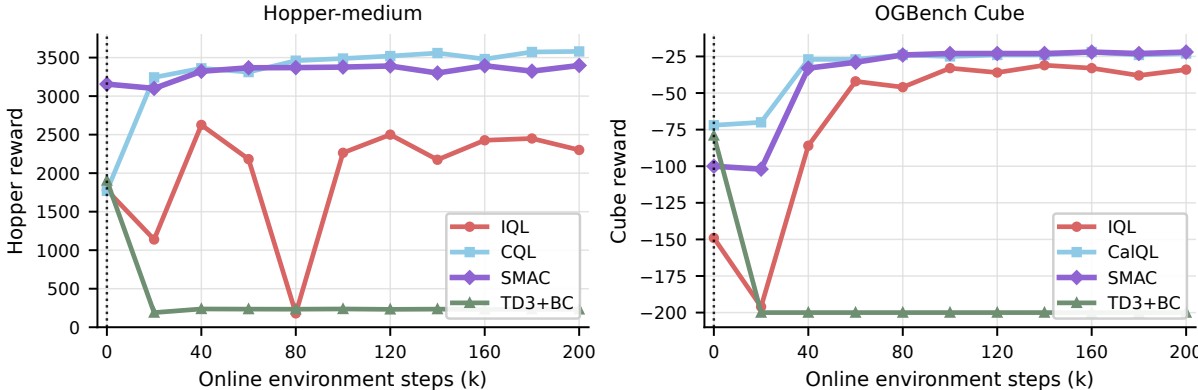

*Figure 17.* **Additional Hopper and Cube transfer results**. We evaluate offline-to-online transfer in Hopper-medium and OGBench Cube. SMAC remains stable in both settings, including Cube where the behavior data is generated by preset motions rather than by an actor-critic policy.

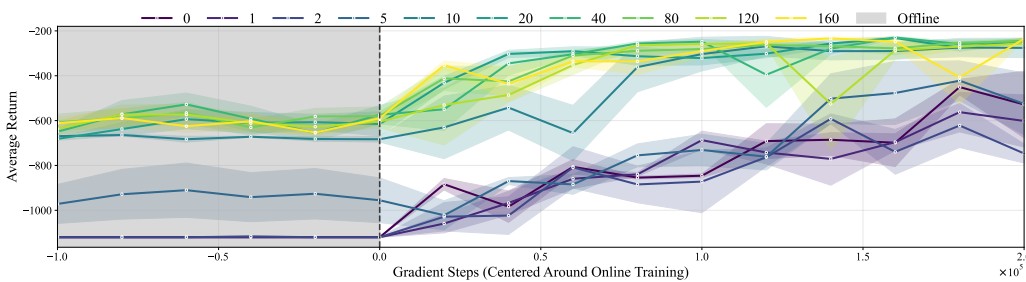

*Figure 18.* **SMAC is robust to varying $\kappa$ once $\kappa$ exceeds a certain threshold**. SMAC shows good performance across a wide range of values for $\kappa$, with performance improving as $\kappa$ increases before eventually levelling off. Once $\kappa > 5$ performance begins to look the same between runs.

model $\epsilon_\theta(x, k)$ estimates $\nabla \log p(x)/\sqrt{\bar{\alpha}_k}$, a scaled version of the score.

### O.1. Hyper-parameters

We use the CleanDiffuser (Dong et al., 2024) package for initializing, training, and inferencing the diffusion models. Within the package we make use of the IDQMLP introduced by Hansen-Estruch et al. (2023). Recall for RvS conditioning the value is min-max normalized during training so 1 is the highest the model has seen. We list the hyper-parameter choices below:

## P. SMAC Hyper-parameters

## Q. Baselines

### Q.1. IQL

IQL learns three networks a critic $Q_\phi$, value $V_\psi$, and policy $\pi_\theta$. The IQL loss functions are as follows:

$$\mathcal{L}(\phi) = \mathbb{E}_{s,a,r,s' \sim \mathcal{D}}[(Q_\phi(s, a) - r - \gamma V_\psi(s'))^2]$$

$$\mathcal{L}(\psi) = \mathbb{E}_{s \sim \mathcal{D}, a' \sim \pi_\theta(a|s)}[|\tau - \delta(V_\psi(s) - Q_\phi(s, a) < 0)|(V_\psi(s) - Q_\phi(s, a))^2]$$

The term $|\tau - \delta(V_\psi(s) - Q_\phi(s, a) < 0)|$ is the expectile loss so differences where $V_\psi(s) - Q_\phi(s, a) > 0$ have $\tau$ weights

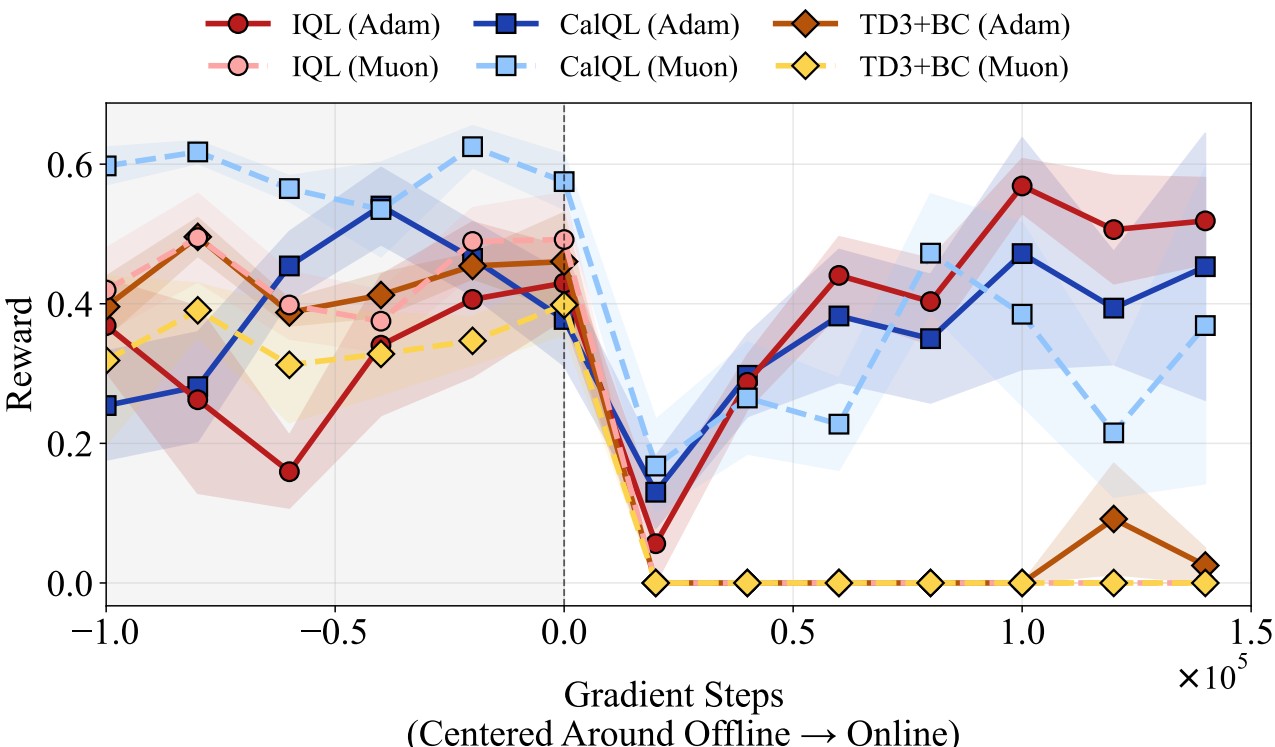

*Figure 19.* **Optimizing the baselines with Muon yields no offline-to-online improvement**

and differences where $V_\psi(s) - Q_\phi(s, a) < 0$ get $(\tau - 1)$ weights. When $\tau > 0.5$ this incentivizes predicting above the mean. The closer $\tau$ is to 1 the more the value network is incentivized to over-shoot the Q-values. The hyper-parameters we use for IQL are listed below

*Table 8.* IQL hyper-parameters across task families (non-diffusion). Door column uses `adroit_iql`. Locomotion/Kitchen columns are filled using the IQL default config in `get_config` and then overridden where specified in the task config.

| Hyper-parameter | Locomotion | Door (Adroit) | Kitchen |
|---|---|---|---|
| Critic learning rate | 0.0003 | 0.0003 | 0.0003 |
| Critic hidden dims | [256, 256] | [256, 256, 256] | [512, 512, 512] |
| Critic activations | relu | relu | relu |
| Critic ensemble size | 2 | 2 | 2 |
| Target update rate | 0.005 | 0.005 | 0.005 |
| Value net learning rate | 0.0003 | 0.0003 | 0.0003 |
| Value hidden dims | [256, 256] | [256, 256, 256] | [512, 512, 512] |
| Value activations | relu | relu | relu |
| Actor optimizer lr | 0.0001 | 0.0001 | 0.0001 |
| Policy std parameterization to be $\geq 0$ | uniform | uniform | uniform |
| Policy hidden dims | [256, 256] | [512, 512, 512] | [512, 512, 512] |
| Policy activations | relu | relu | relu |
| Policy tanh transform distribution | False | False | False |
| Discount $\gamma$ | 0.99 | 0.99 | 0.99 |
| Expectile $\tau$ | 0.9 | 0.7 | 0.7 |
| AWR temperature $\beta$ | 1.0 | 0.5 | 0.5 |

| Hyper-parameter | Locomotion | Door (Adroit) | Kitchen |
|---|---|---|---|
| TD3+BC loss weight $\beta$ | 1.0 | 5.0 | 2.5 |

## Q.2. CalQL/CQL

Calibrated Q-Learning and Conservative Q-Learning are two offline RL methods which use pessimism. Letting $Q_\psi$ be the critic, $\pi_\theta$ the actor and $V^{MC}(s)$ be the observed monte-carlo return for a state $s$ in the dataset the Calibrated Q-Learning loss functions are:

$$\mathcal{L}(\psi) = \mathbb{E}_{\substack{s,a,r,s'\sim\mathcal{D} \\ a'\sim\pi_\theta(a|s)}}[(Q_\phi(s,a) - r - \gamma Q_{\bar\phi}(s',a'))^2] + \alpha(\mathbb{E}_{\substack{s\sim\mathcal{D} \\ a\sim\mathcal{B}(s)}}[\min(V^{MC}(s), Q_\psi(s,a))] - \mathbb{E}_{s,a\sim\mathcal{D}}[Q_\psi(s,a)])$$

$$\mathcal{L}(\theta) = -\mathbb{E}_{s\sim\mathcal{D}, a\sim\pi_\theta(a|s)}[Q_\psi(s,a) - \log(\pi(a|s))]$$

where $\mathcal{B}(s)$ is defined the same as its defined in section 6.

The Conservative Q-Learning loss functions are:

$$\mathcal{L}(\psi) = \mathbb{E}_{\substack{s,a,r,s'\sim\mathcal{D} \\ a'\sim\pi_\theta(a|s)}}[(Q_\phi(s,a) - r - \gamma Q_{\bar\phi}(s',a'))^2] + \alpha(\mathbb{E}_{\substack{s\sim\mathcal{D} \\ a\sim\mathcal{B}(s)}}[Q_\psi(s,a)] - \mathbb{E}_{s,a\sim\mathcal{D}}[Q_\psi(s,a)])$$

$$\mathcal{L}(\theta) = -\mathbb{E}_{s\sim\mathcal{D}, a\sim\pi_\theta(a|s)}[Q_\psi(s,a) - \log(\pi(a|s))]$$

From this it becomes clear how CQL is the antecedent algorithm to CalQL as CalQL modifies the second term on the CQL objective to mitigate under-estimation. In the table below the Locomotion column is for CQL while the other columns show hyper-params for CaQL

*Table 9.* CQL / Cal-QL hyper-parameters across task families

| Hyper-parameter | Locomotion | Adroit | Kitchen |
|---|---|---|---|
| Critic learning rate | 0.0003 | 0.0003 | 0.0003 |
| Critic hidden dims | [512, 512, 512] | [512, 512, 512, 512] | [512, 512, 512] |
| Critic activations | relu | relu | relu |
| Critic ensemble size | 10 | 10 | 10 |
| Target update rate | 0.005 | 0.005 | 0.005 |
| Policy tanh-squash dist. | True | True | True |
| Policy std parameterization to be $\geq 0$ | exp | exp | exp |
| Policy hidden dims | [512, 512] | [256, 256, 256] | [512, 512, 512] |
| Policy activations | relu | relu | relu |
| Discount $\gamma$ | 0.99 | 0.99 | 0.99 |
| Offline batch size | 64 | 64 | 64 |
| Online batch size | 512 | 512 | 512 |
| Offline Gradient steps | 250,000 | 200,000 | 400,000 |
| CalQL/CQL $\alpha$ (cql_alpha) | 5.0 | 5.0 | 5.0 |
| AWR temperature $\beta$ | 1.0 | 0.5 | 1.0 |
| TD3+BC loss weight $\beta$ | 2.5 | 2.5 | 5.0 |
| SAC target entropy | $-10 \cdot |\mathcal{A}|$ | $-10 \cdot |\mathcal{A}|$ | $-10 \cdot |\mathcal{A}|$ |

## Q.3. TD3+BC

The TD3+BC (Fujimoto & Gu, 2021) loss functions are as follows for a critic parameterized by $\phi$ with target params $\bar\phi$, and policy parameterized by $\theta$

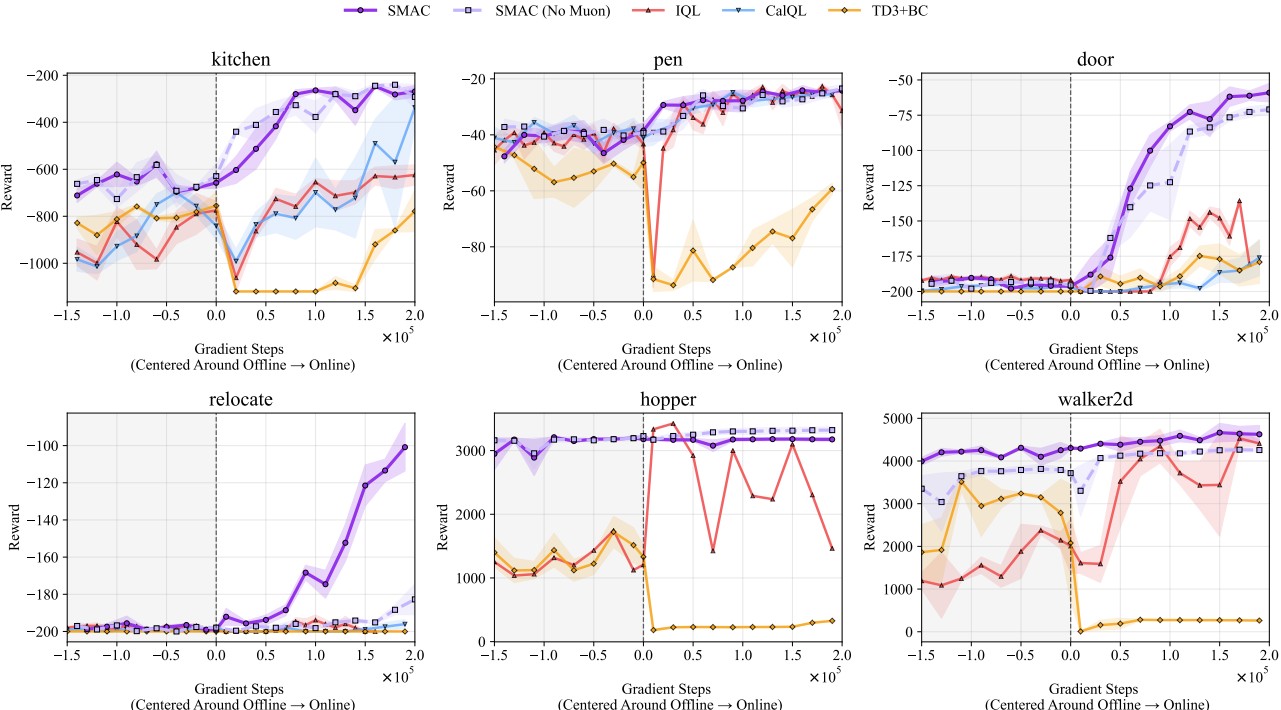

*Figure 20.* **Muon critical for offline-to-online success of SMAC** we plot the offline-to-online curves for SMAC when using Muon vs. Adam optimizers. We observe that using Adam leads to a drop in performance during transfer in 3/6 environments whereas using Muon experiences no drop across all environments.

$$\mathcal{L}(\phi) = \mathbb{E}_{\substack{s,a,r,s'\sim\mathcal{D} \\ a'\sim\pi_\theta(a|s)}}[(Q_\phi(s,a) - r - \gamma Q_{\bar{\phi}}(s',a'))^2$$

$$\mathcal{L}(\theta) = -\mathbb{E}_{s,a\sim\mathcal{D},a'\sim\pi_\theta(a|s)}\Big[-\frac{Q_\phi(s,a)}{sg(|Q_\phi(s,a)|)} + \beta||a'-a||_2^2\Big]$$

where $sg(\cdot)$ is the stop-gradient operator. We use the following hyper-parameters for TD3+BC in each environment:

*Table 10.* TD3+BC hyper-parameters across task.

| Hyper-parameter | Locomotion | Adroit | Kitchen |
|---|---|---|---|
| Critic learning rate | 0.0003 | 0.0003 | 0.0003 |
| Critic hidden dims | [512, 512, 512] | [512, 512] | [512, 512, 512] |
| Critic activations | relu | relu | relu |
| Critic ensemble size | 10 | 10 | 10 |
| Critic target update ratio | 0.005 | 0.005 | 0.005 |
| Policy learning rate | 0.0001 | 0.0001 | 0.0001 |
| Policy hidden dims | [512, 512] | [512, 512] | [512, 512, 512] |
| Policy std transformation to be $\geq 0$ | exp | exp | exp |
| Policy activations | relu | relu | relu |
| Discount $\gamma$ | 0.99 | 0.99 | 0.99 |
| Offline batch size | 32 | 32 | 32 |
| Online batch size | 1024 | 1024 | 1024 |
| Offline Gradient Steps | 250,000 | 200,000 | 400,000 |
| TD3 BC loss weight $\beta$ | 2 | 15 | 5.0 |

| | | | |
|---|---|---|---|
| AWR Temperature $\beta$ | 1 | 1 | 1 |
| Sac target entropy | $-|\mathcal{A}|$ | $-|\mathcal{A}|$ | $-|\mathcal{A}|$ |

## R. Generating contour graphs of planes in parameter space

To generate and plot a plane in parameter space from three parameterizations $\theta_1, \theta_2, \theta_3$ we follow Mirzadeh et al. (2020) and do the following: get difference vectors $u := \theta_2 - \theta_1, v := \theta_3 - \theta_1$ which span the plane; orthogonalize them by making $u' := u, v' := v - cos(u, v)v$; sample parameters along grid of parameters defined by $\theta(t, l) = \theta + lu + tv$ where $t, l \in [-0.2, 1.2] \times [-0.2, 1.2]$ and run in environment to get corresponding reward. Following this we use matplotlib's `contour` plotting function to generate the plot from the grid data.

## S. Experimental Details

### S.1. Benchmark Environments

The `hopper-medium-replay-v2` and `walker2d-medium-replay-v2` were chosen as they are datasets with purely suboptimal behaviours from which optimal behaviour can be learned. `kitchen-partial-v0` was picked as it is an example of a long-horizon task that involves composing the completion of 4 independent tasks together. Lastly, `door-binary-v0`, `relocate-binary-v0`, and `pen-binary-v0` were picked as they are sparse reward, long-horizon, and high-dimensional tasks where offline RL methods are known to struggle. The modifications made to them are removing failure demonstrations, and terminating episodes at states where the goal is achieved which changes the termination condition of the environment.

### S.2. A note on Offline-to-Online Specific Benchmarks

We choose not to use online RL algorithms like O3F SUNG (Guo et al., 2024), PROTO (Li et al., 2023), or PEX (Zhang et al., 2023a) specifically designed for offline-to-online comparison because they are generally working on the inverse of our problem statement. Finding one online RL algorithm which works for as many offline RL algorithms. We show that sufficient BC regularization of the policy in the online phase is generally sufficient for this to be accomplished. Additionally, several of these works demonstrate significantly longer online training periods in the millions of steps to accomplish rewards which we report in the first few thousand steps when fine-tuning with SAC or TD3+BC.

### S.3. Evaluating CalQL vs CQL

In infinite-horizon environments datasets fail to have full Monte-Carlo returns since trajectories must be truncated to fit into a finite dataset. So, in datasets where Monte-Carlo returns are available, we use CalQL, and when Monte-Carlo returns are unavailable we use CQL. The two environments which don't have Monte-Carlo returns in their datasets that we test on are `hopper-medium-replay-v2` and `walker2d-medium-replay-v2` as they are infinite-horizon tasks.

## T. Algorithm Pseudocode

We present our algorithm Score Matched Actor-Critic (SMAC) in pseudocode. We point out that SMAC is only defined for the offline phase since our intent is for SMAC to be usable by arbitrary online actor-critic algorithms.

## U. SAC and PPO respect identity at Optima

### U.1. SAC

The SAC loss function for a policy $\pi$ with Q-function $Q^{\pi_{old}}$ is

By assumption, let $\nabla_a \log(\pi(a|s)) = \nabla_a Q(s, a)$. Then since $\pi(a|s)$ must satisfy the axioms of probability measures we have $\int_A \pi(a|s) = 1$, hence. Now, keeping $s$ fixed we consider $Q$ and $\pi$ along only $a$.

$$\nabla_a \log(\pi(a|s)) = \nabla_a Q(s, a)$$

Then integrating with respect to $a$ gives

*Table 6.* Diffusion-model hyper-parameters used by SMAC (diffusion BC) across task families. Each column corresponds to the diffusion configuration used for that environment family.

| Hyper-parameter | Locomotion | Adroit Tasks | Kitchen |
|---|---|---|---|
| Obs Dimension / Emb Dimension | 64 / 64 | 64 / 64 | 64 / 64 |
| Hidden Dimension | 512 | 512 | 512 |
| # of blocks | 3 | 5 | 5 |
| Condition Network architecture | MLP | MLP | MLP |
| Condition Network Hidden Dimensions | [512, 512, 512] | [256, 256, 256] | [256, 256, 256] |
| Condition out dim | 64 | 64 | 64 |
| Diffusion steps ($K$) | 32 | 32 | 32 |
| Training steps | 1,000,000 | 3,000,000 | 3,000,000 |
| Batch size | 512 | 1024 | 1024 |
| RvS Conditioner | Trajectory Reward | Trajectory Reward | Number of Tasks Completed |
| RvS inference value $\gamma_{\text{rtg}}$ | 1.0 | 1.0 | 1.0 |

---

**Algorithm 1** Score Matched Actor-Critic: Offline Phase

---

**Require:** Learning rates $\delta_Q, \delta_\pi, \delta_\alpha$ and Polyak term $\lambda_Q$
  Pre-train $\epsilon_\omega$ with RvS and the Diffusion Loss
  initialize $Q_\theta, \pi_\phi, \alpha_\psi$
  initialize $Q_{\bar\theta}$ with $\bar\theta = \theta$
  **for** $N$ offline steps **do**
    Sample batch $\{(s_i, a_i, r_i, s_i')\} \sim \mathcal{D}$
    Sample actions $\bar{a}_i$ for score-matching loss
    $\theta \leftarrow \theta - \delta_Q \text{Muon}(\nabla_\theta \mathcal{L}^{SMAC}(\theta, \psi))$
    $\psi \leftarrow \psi - \delta_\omega \text{Muon}(\nabla_\psi \mathcal{L}^{SMAC}(\theta, \psi))$
    $\phi \leftarrow \phi - \delta_\pi \text{Muon}(\nabla_\phi \mathcal{L}^\pi(\phi))$
    $\bar\theta \leftarrow \lambda_Q \theta + (1 - \lambda_Q)\bar\theta$
  **end for**

---

$$\log(\pi(a|s)) = \frac{1}{\alpha}Q(s,a) + C$$

Now, exponentiating we arrive at

$$\pi(a|s) = exp(\frac{1}{\alpha}Q(s,a) + C)$$

$$\pi(a|s) = exp(C)exp(\frac{1}{\alpha}Q(s,a))$$

But, since $\int_A \pi(a|s) = 1$ we must have $exp(C) = \frac{1}{\int_A \frac{1}{\alpha}Q(s,a)} = \frac{1}{Z(s)}$ and hence:

$$\pi(a|s) = \frac{exp(\frac{1}{a}(Q(s,a))}{Z(s)}$$

But since the SAC policy loss is:

$$D_{KL}(\pi(a|s)||\frac{exp(\frac{1}{\alpha}Q(s,a))}{Z(s)})$$

We have that $\pi$ minimizes the loss.

*Table 7.* SMAC (non-diffusion) hyper-parameters across task families. Diffusion-related fields are excluded and in section above.

| Hyper-parameter | Locomotion | Adroit Tasks | Kitchen |
|---|---|---|---|
| Critic Learning rate | 0.0003 | 0.0003 | 0.0003 |
| Critic hidden dims | [512, 512, 512] | [512, 512, 512, 512] | [512, 512, 512, 512] |
| Critic activations | tanh | tanh | tanh |
| Critic ensemble size | 10 | 10 | 10 |
| Critic Target Update Ratio | 0.005 | 0.005 | 0.005 |
| Policy Learning rate | 0.0001 | 0.0001 | 0.0001 |
| Policy std transform to be $\geq 0$ | exp | exp | exp |
| Policy hidden dims | [512, 512, 512] | [256, 256, 256] | [512, 512, 512] |
| Policy activations | relu | relu | relu |
| $\alpha(s)$ optimizer learning rate | 0.0001 | 0.0001 | 0.0001 |
| $\alpha(s)$ Hidden Dimensions | [256, 256] | [256, 256] | [256, 256] |
| $\alpha(s)$ Activation | relu | relu | relu |
| $\kappa$ | 40 | 40 | 50 |
| Discount $\gamma$ | 0.99 | 0.99 | 0.99 |
| Offline Batch Size | 64 | 64 | 64 |
| Online Batch Size | 1024 | 1024 | 1024 |
| Offline Gradient Steps | 250,000 | 200,000 | 400,000 |
| AWR Temperature | 0.4 | 20.0 | 5.0 |
| TD3+BC BC loss weight | 5.0 | 0.2 | 2 |
| SAC Target Entropy | $-10\cdot|\mathcal{A}|$ | $-10\cdot|\mathcal{A}|$ | $-10\cdot|\mathcal{A}|$ |

## U.2. PPO

At convergence $\pi_{\theta_{old}} = \pi_\theta$, then for a fixed state $s$ and data collected by $\pi$ PPO gets the policy to maximize this expression

$$= \mathbb{E}_{a\sim\pi_{\theta_{old}}(a|s)}[A(s,a)\frac{\pi_\theta(a|s)}{\pi_{old}(a|s)}] - c_2\mathbb{E}_{a\sim\pi_\theta(a|s)}[\log(\pi_\theta(a|s))]$$

Recognize that $\theta_{old}$ and divide out by $\frac{1}{c_2}$:

$$= \mathbb{E}_{a\sim\pi_\theta(a|s)}[\frac{1}{c_2}A(s,a)] - \mathbb{E}_{a\sim\pi_\theta(a|s)}[\log(\pi(a|s))]$$

Apply definition of advantage to separate into $Q$ and $V$:

$$= -\mathbb{E}_{a\sim\pi_\theta(a|s)}[\frac{1}{c_2}Q(s,a)] + \frac{1}{c_2}V(s) - \mathbb{E}_{a\sim\pi_\theta(a|s)}[\log(\pi(a|s))]$$

Apply $log(exp(\cdot))$ to Q

$$= \mathbb{E}_{a\sim\pi_\theta(a|s)}[\log(\exp(\frac{1}{c_2}Q)))] - \mathbb{E}_{a\sim\pi_\theta(a|s)}[\log(\pi(a|s))] + V(s)$$

Apply log rules unify the two left terms into one fraction

$$= \mathbb{E}_{a\sim\pi_\theta(a|s)}[\log(\frac{exp(\frac{1}{c_2}Q))}{\pi(a|s))})] + \frac{1}{c_2}V(s)$$

Raise the first term to $((\cdot)^{-1})^{-1}$ and applying log rules bring one exponent to the front and apply the other to flip the fraction.

$$= -\mathbb{E}_{a\sim\pi_\theta(a|s)}[\log(\frac{\pi(a|s)}{exp(\frac{1}{c_2}Q)})] + V(s)$$

add and subtract $log(Z(s))$:

$$= -\mathbb{E}_{a \sim \pi_\theta(a|s)}[\log(\frac{\pi(a|s)}{exp(\frac{1}{c_2}Q)}Z(s))] + V(s) - log(Z(s))$$

bring the $Z(s)$ down to the denominator

$$= -\mathbb{E}_{a \sim \pi_\theta(a|s)}[\log(\frac{\pi(a|s)}{\frac{exp(\frac{1}{c_2}Q)}{Z(s)}})] + V(s) - log(Z(s))$$

and observe that the first term is a KL-divergence:

$$= -D_{KL}(\pi(a|s)||\frac{exp(\frac{1}{c_2}Q)}{Z(s)}) + V(s) - Z(s)$$

By the proof in the section above, we see then that $\pi$ maximizes this objective function since $V(s)$ and $Z(s)$ are invariant to it. For the first update step of line 6 in the original PPO algorithm (Schulman et al., 2017), $\theta = \theta_{old}$. So, this objective above matches the objective at the start of the training part for every iteration of the PPO algorithm. If $\nabla_a \log(\pi(a|s)) = \frac{1}{c_2}\nabla_a Q(s, a)$ then the objective above is maximized. Since the objective above matches the objective at the first step in the training section of the PPO loop the policy will not be updated. Then $\theta = \theta_{old}$ for the following updates in the training section and resulting iterations through the PPO algorithm. Thus, $\theta$ maximizes the objective.

