# OpenReview forum: "SMAC: Score-Matched Actor-Critics for Robust Offline-to-Online Transfer"
_ICML.cc/2026/Conference — ICML 2026 regular_

### Official Review · Reviewer_gCgX · 2026-03-02

**Soundness:** 4
**Presentation:** 4
**Significance:** 3
**Originality:** 3
**Overall Recommendation:** 5
**Confidence:** 4

**Summary:**

While offline actor-critic RL can find good solutions, common methods to fine-tune the model on online RL often experience an early drop in reward. The paper hypothesizes that this is due to reward valleys between the offline RL solution and an online maximum, caused by a misalignment between offline and online RL methods. This is demonstrated by mapping a linear interpolation in parameter space between pretrained (offline) and fine-tuned (online) agent. A solution is proposed: Score Matched Actor-Critic, an online RL method that improves Soft Actor-Critic by using critic regularization and the Muon optimizer to find solutions that are better connected to online solutions.

**Compliance With Llm Reviewing Policy:**

Affirmed.

**Final Justification:**

The rebuttal and further author response addressed my concerns and I have raised my score as indicated in response.

**Key Questions For Authors:**

1. What are the implications of non-monotonically increasing reward in practice? If a high reward is achieved by the end of training, and the training is stable, how much does the path it took matter?

**Limitations:**

yes

**Strengths And Weaknesses:**

Strengths:
1. Good demonstration of the problem on non-monotonically increasing reward during fine-tuning.
2. The empirical results largely show improvement from SMAC pre-training into a variety of online algorithms, compared to other offline RL algorithms.
3. Additional results in the appendix, including the ablation, strengthen the claims made.

Weaknesses:
1. SMAC did not show improvements for the MuJoCo environments (hopper and walker2d) during fine-tuning.
2. For some of the fine-tuning experiments, the model does not appear to show the reward of the initialized models. This makes it appear that the SMAC model is just initialized better for kitchen and MuJoCo than the others rather than demonstrating clear gains in early offline learning.
3. The increased computation cost for the diffusion model is noted as significant in the limitations, but there is no analysis of the costs. This also makes the comparisons between other offline RL methods a bit less fair, as this cost difference is not clear in the results figures, and the methods are compared by offline and online RL optimization steps, while SMAC requires pre-offline training steps for the difffusion model.

Overall, the method and results are good, but their significance is impacted by non-trivial increases in computational costs compared to competing methods. The proposed algorithm is designed to combat the problem of non-monotonically increasing reward for fine-tuning offline-to-online RL, but the paper has not convinced me that this is an issue (see Question 1), given that other methods can achieve high reward after an early fine-tuning reward drop.

---

> ### Author Rebuttal · Authors · 2026-03-31
>
> We would like to thank the reviewer for their detailed comments. We appreciate that the reviewer found our presentation to be excellent, saying that we demonstrated the problem well, and our empirical results and appendices strengthened our claims
>
> # (Weakness 1)
>
> Our primary focus in this work is on transfer stability, specifically, avoiding the initial performance collapse when switching to online fine-tuning and maintaining a low regret. While the fine-tuning does not increase performance in Hopper to the level of the best observed policy, it still remains near-optimal. Additionally, we observe a slight monotonic increase in the Walker2d environment. Altogether, we still consider the MuJoCo results to be strong, as we observe from the plot and Tables in Appendix G that SMAC in MuJoCo environments attains strong offline performance, avoids drops, and achieves the second lowest regret in the MuJoCo environments.
>
> #  (Weakness 2)
>
> In the experiments that compare the offline checkpoint to the finetuned checkpoint, all models are taken after 300k gradient steps. The nature of Offline RL is that different algorithms will have different final performance, and it is part of our argument that the *characteristics of maxima, not just their performance, are crucial for offline-to-online performance*. In the table below, we show experiments in the Door task. We increase the dataset sizes to contain over 10 million samples so that the dataset possesses a level of coverage and quality that allows algorithms to achieve near-optimal and similar offline performances to one another. Nonetheless, when we transfer these optimally offline-trained actor-critics, we observe that the other algorithms experience a sharp drop before recovering to optimal performance. This counters the idea that performance is the criterion, but that there are underlying differences in the actor-critics found by the different regularizations across methods.
>
> ### Algorithms in Door with 10M dataset
> |  | 0 | 20000 | 40000 | 60000 | 80000 | 100000 | 120000 | 140000 | 160000 | 180000 | 200000 |
> |---|---|---|---|---|---|---|---|---|---|---|---|
> | IQL | -88.69 | -125.00 | -56.27 | -55.63 | -51.58 | -51.41 | -54.08 | -51.90 | -51.28 | -50.30 | -51.30 |
> | CalQL | -76.69 | -126.79 | -159.76 | -128.24 | -123.67 | -124.19 | -112.97 | -101.98 | -107.87 | -111.92 | -97.48 |
> | SMAC | -80.43 | -51.26 | -43.00 | -42.41 | -41.10 | -40.62 | -40.03 | -39.39 | -40.03 | -39.33 | -38.70 |
> | TD3+BC | -97.21 | -193.76 | -197.39 | -186.64 | -187.83 | -193.32 | -195.85 | -190.21 | -191.26 | -193.57 | -182.12 |
>
>
> # (Weakness 3)
>
> We provide the following breakdown: The diffusion model for score estimation is trained once on the static dataset, which adds approximately 3-4 GPU hours of pre-training compute on a single L40S. The online RL phases themselves run at a comparable cost to the baselines. The end-to-end wall-clock time for SMAC's offline phase is approximately 3x that of IQL and 2x that of Cal-QL on the same hardware. We present a table that shows regret in the Kitchen environment when training IQL for 3x gradient steps and CalQL for 2x gradient steps to compare, giving the two most promising baselines similar amounts of time. However, our original results test offline-to-online once the runs had converged, so we see no significant changes from the results presented in the paper.
> We acknowledge this overhead and will add this table to the appendix.
>
>
> | Group | 0 | 20000 | 40000 | 60000 | 80000 | 100000 | 120000 | 140000 | 160000 | 180000 | 200000 |
> |-------|------|------|------|------|------|------|------|------|------|------|------|
> | CalQL Wallclock equivalent offline stage | -627.54 | -863.72 | -775.52 | -951.55 | -845.24 | -724.76 | -617.89 | -621.40 | -665.74 | -654.11 | -662.16 |
> | IQL Wallclock equivalent offline stage | -669.83 | -771.73 | -725.86 | -804.34 | -717.52 | -665.77 | -693.90 | -684.76 | -666.90 | -734.06 | -746.99 |
> | SMAC -> SAC | -657.61 | -602.95 | -512.98 | -416.96 | -280.19 | -264.38 | -278.65 | -348.74 | -246.75 | -281.77 | -268.94 |
>
>
>
> # Key Questions For Authors:
>
> We believe the path matters significantly in three concrete scenarios: (1) Early stopping and sample budgets, when online environment interaction is expensive, dangerous, or time-limited. A method that drops sharply before recovering can fail to meet thresholds within the available budget, even if it would eventually recover given unlimited steps; (2)
> Safety-critical and real-world deployment, any domain where the agent operates in the real world during fine-tuning, a performance drop means the agent is executing worse behavior on a physical system during the recovery period, This can cause damage, increase risk, or invalidate the case for online fine-tuning entirely; (3) Warm-starting sample-efficient algorithms, one of the motivations for offline pre-training is to provide a warm start for online RL. If fine-tuning causes an immediate drop, the warm start is negated.

---

> > ### Author Rebuttal · Reviewer_gCgX · 2026-03-31
> >
> > The additional results provided demonstrate the generalizability of the algorithm and that the results do not rely on the increased algorithmic costs of the method. The answer to question 1 justifies why pure performance is not the ultimate goal. I suggest that the authors add some of the question 1 response to the introduction: this will help clarify the motivation and the significance of the results.
> >
> > I have updated my score. I increased soundness due to the additional results provided and to the clarification on costs. I increased significance due to the clarified motivation and use cases. To reflect this, I increased the overall recommendation and my confidence.
> >
> > (RESOLVED)
> > The following is a minor issue and won't affect my final score, but I think clearing this up could make the results more interpretable with minimal changes to the manuscript.
> >
> > My remaining issue alongside weakness 2 is more with the presentation of the results, rather than the actual algorithm or results. The graphs in Figures 5-7 appear to start at -100k epochs, but you stated here that you do 300k epochs of offline RL. Is that correct? If I'm not misunderstanding, this means the first 200k offline RL epochs are not shown. This isn't a major issue, but it isn't stated in the figure captions. Clarifying in the captions that the graph does not contain all offline RL gradient steps could be sufficient to clear up some confusion if adding the missing 200k epochs to the graphs is not feasible.

---

> > > ### Author Response · Authors · 2026-04-01
> > >
> > > We're glad that our rebuttal was well received and has resolved the main concerns/questions. The understanding that we omit the first 200k offline steps is correct. We include only the offline steps to contrast how gradient steps at the start of the online setting affect performance with the final gradient steps in the offline setting. We will clarify this in the caption to remove any future confusion.
> > >
> > > Thank you,
> > > Authors

---

### Official Review · Reviewer_cYgf · 2026-03-04

**Soundness:** 2
**Presentation:** 3
**Significance:** 3
**Originality:** 3
**Overall Recommendation:** 4
**Confidence:** 4

**Summary:**

This paper proposes Score-Matched Actor-Critics (SMAC), an offline reinforcement learning algorithm designed to mitigate the severe performance degradation typically observed at the start of online RL fine-tuning. The authors propose a geometric hypothesis for this degradation: existing offline RL methods converge to local maxima that are separated from the true online optimal policy by low-reward valleys. To traverse this valley, SMAC regularize the Q function during the offline training phase that aligns the action-gradient of the Q-function with the score function of the behavioral policy. Additionally, the method replaces the Adam optimizer with the Muon optimizer. Empirical evaluations on six D4RL datasets suggest that SMAC effectively prevents early-stage degradation during online fine-tuning.

**Compliance With Llm Reviewing Policy:**

Affirmed.

**Final Justification:**

My primary concerns have largely been addressed during rebuttal. I would like to encourage the authors to include more experiments to further validate the effectiveness of SMAC.

**Key Questions For Authors:**

1. While the geometric motivation is clear, the exact mechanics of why matching the Q-function's action gradient to the dataset policy score successfully bridges the low-reward valley is not fully established. Could the authors provide a more detailed explanation?

**Limitations:**

yes

**Strengths And Weaknesses:**

**Strengths**

1. The observation and characterization of "low-reward valleys" separating offline optima from online optima provides a compelling geometric explanation for a well-known empirical problem in offline-to-online RL.
1.  The learning curves presented (e.g., Figures 6 and 7) demonstrate a clear mitigation of the initial fine-tuning degradation, aligning tightly with the paper's primary motivation.

**Weaknesses**

1. The evaluation is critically limited, testing only 6 datasets from the D4RL benchmark. Given the diversity of dynamics and dataset qualities in D4RL, evaluating on only 6 environments is strictly inadequate to establish SMAC as a robust offline-to-online algorithm. The authors should expand their evaluation across a wider variety of D4RL domains.
1. The visualizations in Figure 2 are compelling and insightful; however, they are limited to a single dataset (kitchen-partial-v0). To assess the robustness and generality of the observed effects, it is essential to examine whether similar phenomena consistently emerge across a broader range of datasets.

---

> ### Author Rebuttal · Authors · 2026-03-31
>
> We are glad that the reviewer found our characterization of low reward valleys separating offline and online maxima to be a compelling explanation for the offline-to-online problem, and that they found the experimental section to show that SMAC accomplishes the paper's main motivation
> # (Weakness 1)
>
> In a table below we provide extended results in hopper-medium-v2, and ogbench's cube-play-singletask-task1, we consider these interesting domains worthy of evaluation because: hopper-medium contains strictly sub-optimal data but no extreme failures meaning that the ability of the online learning to explore and improve without falling into a failure mode is especially tested; and ogbench-cube because its behaviour policy is collected by executing pre-set motions many of which have nothing to do with the task or are counter-productive to the task breaking an implicit assumption that the behaviour policy be from an actor-critic. The success of SMAC here shows that even when the data is farther from the assumptions that motivate the regularization, the algorithm still works as intended.
>
> ## Hopper-medium reward
> Algorithm | 0 | 20000 | 40000 | 60000 | 80000 | 100000 | 120000 | 140000 | 160000 | 180000 | 200000 |
> |---|---|---|---|---|---|---|---|---|---|---|---|
> | IQL | 1780 | 1139 | 2626 | 2182 | 180 | 2264 | 2500 | 2174 | 2427 | 2450 | 2301 |
> | CQL | 1771 | 3243 | 3361 | 3312 | 3460 | 3488 | 3520 | 3559 | 3481 | 3573 | 3580 |
> | SMAC | 3156 | 3078 | 3322 | 3368 | 3370 | 3377 | 3391 | 3300 | 3394 | 3324 | 3398 |
> | TD3+BC | 1904 | 190 | 237 | 235 | 234 | 237 | 232 | 235 | 231 | 233 | 232 |
>
> ## Cube reward
> | Algorithm | 0 | 20000 | 40000 | 60000 | 80000 | 100000 | 120000 | 140000 | 160000 | 180000 | 200000 |
> |---|---|---|---|---|---|---|---|---|---|---|---|
> | IQL | -149 | -196 | -86 | -42 | -46 | -33 | -36 | -31 | -33 | -38 | -34 |
> | CalQL | -72 | -70 | -27 | -27 | -24 | -25 | -24 | -24 | -22 | -24 | -23 |
> | SMAC | -100 | -111 | -33 | -29 | -24 | -23 | -23 | -23 | -22 | -23 | -22 |
> | TD3+BC | -79 | -200 | -200 | -200 | -200 | -200 | -200 | -200 | -200 | -200 | -200 |
>
>
> # (Weakness 2)
>
>
> We appreciate this concern. Generating the parameter-space visualizations analogous to Figure 2 across additional environments requires evaluating reward along a dense grid of parameter interpolations for each seed and environment, which is computationally prohibitive within the rebuttal period. In Figure 3 of the paper, we already show that the presence or absence of a valley in the interpolation between offline and online checkpoints corresponds precisely to the presence or absence of a drop in the learning curves across Kitchen, Door, and Hopper; extending the analysis to more than just one environment. Given this established correspondence, the learning curves in Figure 5 across the three environments serve as strong evidence that the valley phenomenon generalizes beyond kitchen-partial. We will include these curves more prominently in the revision with explicit commentary connecting them to the connectivity hypothesis, and will add the parameter-space visualizations for additional environments in the final version, time permitting. We did have time to generate the linear connectivity plots from Figure 3 for CalQL->CalQL and IQL->IQL, and you can see tables showing the data in our reply to reviewer MaEz. The tables support our argument that linear connectivity is consistent with offline-to-online drop.
>
>
>
> # Key Questions:
>
>
> The core motivation of SMAC is our observation in Section 5 that the fundamental problem in offline-to-online transfer is a mismatch between the objectives that offline and online RL optimize. SMAC's regularizer is designed to close this gap by finding offline actor-critics whose stable points are also stable points of online RL.
> The motivating arguments in Section 6, the proofs in Appendix P, and the empirical verification in Appendix F all serve to argue that during online training, actor-critic algorithms are implicitly pushed toward satisfying $\nabla_a \log \pi(a|s)\propto \nabla_a Q(s,a)$. Hence, if offline regularization explicitly optimizes for this property, we obtain actor-critics that share structural characteristics with those found during online training, and thereby avoid a mismatch in optimization stable points. In this way, using $\nabla_a \log \pi_D(a|s)$ as a proxy for $\nabla_a Q^D(s,a)$ ensures that our critic (and the actor it implies through the actor loss) is close to an online-trained actor-critic that could have generated the data. This closeness to an online training trajectory is what gives SMAC's pre-trained actor-critic the roughly monotonic improvement characteristic of online value-based RL.
>
> We will articulate this explanation as a dedicated paragraph in Section 6 of the revision, making the mechanistic connection between the regularizer, the geometry of the Q-function, and the smooth transfer property explicit for the reader.

---

> > ### Author Rebuttal · Reviewer_cYgf · 2026-03-31
> >
> > My primary concerns have largely been addressed. I would like to encourage the authors to include more experiments to further validate the effectiveness of SMAC. I have raised my score accordingly.

---

> > > ### Author Response · Authors · 2026-04-01
> > >
> > > We thank the reviewer for their positive response to our rebuttal and for raising their score. We appreciate the suggestion to include more experiments and will look to incorporate additional experiments validating SMAC where feasible within the final version.
> > >
> > >
> > > Thank you,
> > > Authors

---

### Official Review · Reviewer_MaEz · 2026-03-06

**Soundness:** 2
**Presentation:** 3
**Significance:** 3
**Originality:** 3
**Overall Recommendation:** 4
**Confidence:** 4

**Summary:**

The paper points out that existing offline RL methods could converge to solutions that are not linearly connected to the ones that SAC fine-tuning them finds. To address this issue, the paper proposes an offline RL algorithm that matches the Q-function gradient with the score of the dataset action distribution. Empirical results on 6 D4RL tasks in the offline-to-online setting are presented to validate the transfer stability and online regret of the proposed method.

**Compliance With Llm Reviewing Policy:**

Affirmed.

**Final Justification:**

The rebuttal addressed my main concerns.

**Key Questions For Authors:**

1. In Figure 2 and 4, it seems that the 3 baseline algorithms (Cal-QL, TD3+BC, IQL) produce solutions that are better connected with TD3+BC solutions rather than SAC ones. Does it mean that it's just a compatibility issue between certain offline and online RL algorithms, rather than the difference in general transfer ability? I am willing to increase my score if the authors could provide additional results, including but not limited to:
	- Learning curve comparisons with IQL -> IQL, Cal-QL -> Cal-QL transfer
	- Parameter-space visualization containing solutions finetuned by IQL and Cal-QL.

**Limitations:**

Yes.

**Strengths And Weaknesses:**

**Strengths**
- The paper empirically analyzes the linear connectivity between solutions found by offline RL pretraining and online SAC/TD3+BC finetuning. To the best of my knowledge, this perspective is novel.
- The paper is clearly written and well organized.

**Weaknesses**
- The proposed score matching regularization method is not well motivated by the overall narrative. The authors claim that it comes from the identity $\nabla_{a} \log \pi^{\*}(a | s) = \frac{1}{\alpha} \nabla_{a} Q^{\*}(s, a)$. However,
	- The behavior policy $\pi^{\mathcal{D}}$ is generally suboptimal, which means that $\pi^{\mathcal{D}} \neq \pi^{\*}$.
	- "if Q is frozen and only the $\pi$ objective is considered": The proposed score matching loss optimizes exactly the $Q$ network, so "Q is frozen" does not hold.
	- "Even when it does not hold, exactly matching ... would result in OOD actions being penalized proportionally to how OOD the action is.": It is not clear (i) how this claim relates to the previous narrative on **mode connectivity**, or (ii) **why** the proposed method converges to a solution that's better connected with the SAC-finetuned solution.
	- Regarding the SAC and PPO argument in Appendix P, (informally) the action gradient of a policy $\pi$ and an action value function $Q(s, a)$ would approximately match because they form an actor-critic relationship. But in general, the learned $Q_{\theta}(s, a)$ is not necessarily the critic of the behavior policy $\pi^{\mathcal{D}}(\cdot | s)$. Therefore, it makes little sense to match their gradients.
- The significance of the problem the paper aims to address could be strengthened. For example, both the visualization and the offline-to-online benchmarks are limited to using SAC, TD3, TD3+BC and AWR as the online algorithm. Since the original Cal-QL and IQL paper uses the same algorithm for both offline and online phase, additional comparisons with the best performing pairs would strengthen the claims (see Key Questions).

---

> ### Author Rebuttal · Authors · 2026-03-31
>
> We would like to thank the reviewer for finding our paper well written and organized in its presentation of a novel perspective
> # Weakness 1
> We agree that our presentation overstated the score-matching identity as a formal derivation. Our claim is not that the dataset policy $\pi^D$ is optimal, that $Q_\theta$ is its critic, or that $\nabla_a \log \pi^D(a\mid s)\propto \nabla_a Q(s, a)$ should hold exactly offline. Rather, the identity is only motivation from online actor-critic optima such as SAC, where policy score and critic action-gradient are aligned.
>
> We also agree that our previous OOD discussion did not clearly connect to mode connectivity. Our main point is that the key issue in offline-to-online transfer is an objective mismatch between offline pessimistic training and online actor-critic optimization. The motivating arguments in Section 6, the proofs in Appendix P, and the empirical verification in Appendix F were meant to argue that during online training, actor-critic algorithms are implicitly pushed toward satisfying $\nabla_a \log \pi(a|s)\propto \nabla_a Q(s, a)$.  In SMAC, $\nabla_a \log \pi^D(a| s)$ is used only as an offline-computable proxy signal that biases the critic (and policy it induces through the policy loss) toward a structure more compatible with an online actor-critic that could have generated the dataset.
> Hence, our offline regularization explicitly optimizes for a property we believe online training implicitly optimizes towards, and thereby avoids a mismatch in optimization stable points. This is the motivation for SMAC.
> # Weakness 2
> The significance of the problem stems from the main argument that offline actor-critics should be able to act as pre-trained models, which can be fine-tuned by an arbitrary family of online RL methods. Evidently, current offline RL methods fail in this regard and suffer a drop in performance, to which we point in our reply to reviewer gCgX's key question for an in-depth explanation of why the drop is undesirable.
>
> One observation in the paper is that using offline algorithms in the fine-tuning phase (TD3+BC, CalQL, IQL, etc.) may allow for monotonic improvement, but at the cost of much higher regret than using a more data-efficient online method; we illustrate this by using TD3+BC for fine-tuning. For completeness, we have provided the offline-to-online results for using CalQL in both phases and IQL in both phases in the Kitchen environment as part of our reply to the key question.
> # Question
> Yes, we attempt to elaborate on this at the end of the experimental section, but we will make it more of a focal point in our final paper. In the tables below, we have the reward (table 1) and interpolation results (table 2) for running a CalQL-\>CalQL and IQL -\> IQL offline-to-online runs. The results mimic our results in the paper, which show that using offline objectives in the online phase results in stable offline-to-online transfer, but at the cost of significantly slower learning. We also present results in a table in our reply to reviewer gCgX that offline-to-online cannot be purely alleviated by increasing the dataset size, coverage, and optimality of the dataset by showing that transfer remains an issue even on large, high-coverage, high-quality datasets. This points towards offline-to-online being a compatibility issue between offline and online optima and motivates SMAC, and its compatibility across a range of different algorithms.
> # Table 1 (reward curves)
> | | 0 | 2000 | 40000 | 60000 | 80000 | 100000 | 120000 | 140000 | 160000 | 180000 | 200000 |
> |-------|------|------|------|------|------|------|------|------|------|------|------|
> | IQL -> IQL | -749 | -738 | -596 | -601 | -610 | -632 | -583 | -604 | -595 | -582 | -579 |
> | CalQL -> CalQL | -639 | -595 | -558 | -545 | -528 | -504 | -501 | -506 | -500 | -502 | -485 |
> | IQL -> SAC | -774 | -1061 | -863 | -725 | -758 | -654 | -713 | -698 | -628 | -633 | -624 |
> | CalQL -> SAC | -840 | -991 | -835 | -788 | -807 | -698 | -771 | -722 | -489 | -569 | -337 |
> | SMAC -> SAC | -657 | -602 | -512 | -416 | -280 | -264 | -278 | -348 | -246 | -281 | -268 |
>
> # Table 2 (interpolations)
> | | -0.13 | -0.09 | -0.05 | -0.01 | 0.03 | 0.07 | 0.11 | 0.15 | 0.19 | 0.23 | 0.28 | 0.32 | 0.36 | 0.40 | 0.44 | 0.48 | 0.52 | 0.56 | 0.60 | 0.64 | 0.68 | 0.72 | 0.77 | 0.81 | 0.85 | 0.89 | 0.93 | 0.97 | 1.01 | 1.05 | 1.09 | 1.13 |
> |---|---|---|---|---|---|---|---|---|---|---|---|---|---|---|---|---|---|---|---|---|---|---|---|---|---|---|---|---|---|---|---|---|
> | calql  | -647 | -667 | -643 | -611 | -623 | -624 | -615 | -647 | -606 | -557 | -608 | -567 | -546 | -554 | -545 | -552 | -527 | -510 | -515 | -504 | -497 | -486 | -479 | -481 | -483 | -483 | -483 | -491 | -490 | -512 | -509 | -507 |
> | iql   | -824 | -809 | -729 | -775 | -735 | -704 | -630 | -685 | -559 | -617 | -586 | -618 | -573 | -599 | -587 | -542 | -550 | -552 | -572 | -603 | -574 | -601 | -569 | -601 | -573 | -591 | -599 | -552 | -588 | -575 | -557 | -612 |

---

> > ### Author Rebuttal · Reviewer_MaEz · 2026-04-01
> >
> > The rebuttal changed my overall assessment of the paper. On Weakness 1, I understand the motivation for the score-matching objective better now. I still think the beginning of Section 6 **should be stated more carefully**, since the exact max-entropy identity does not directly justify matching the critic action gradient to the dataset score in the way the current paragraph may suggest. That said, I do not view this as a blocking issue, and I think the empirical analysis and offline-to-online results are strong enough to outweigh it.
> >
> > On Weakness 2 and my key question, the additional results were helpful. They clarify that using the same offline algorithm in the online phase can give smoother transfer, but SAC fine-tuning performs better, or at least on par, overall. This makes the choice of SAC in the online phase well justified and addresses my earlier concern that the effect might mainly be due to compatibility between particular offline and online algorithms.
> >
> > Overall, the rebuttal was enough to move me toward a more positive recommendation. I have therefore increased my significance score to 3 and my overall recommendation to 4.

---

> > > ### Author Response · Authors · 2026-04-01
> > >
> > > We would like to thank the reviewer for their original review and thoughtful response to our rebuttal. We will rewrite the beginning of section 6 to more closely reflect our response. We will also include the results we have shared in the rebuttal in the final version.
> > >
> > > Thank you,
> > > Authors

---

### Official Review · Reviewer_2fpa · 2026-03-11

**Soundness:** 3
**Presentation:** 1
**Significance:** 3
**Originality:** 3
**Overall Recommendation:** 5
**Confidence:** 3

**Summary:**

This paper studies a setting where models in reinforcement learning are pre-trained on offline data and then fine-tuned online. The work studies the drop in performance that occurs when switching from the offline to online phase and attributes it to non-linear return relationships in parameter space between pre-trained and final solutions. This is generally studied as linear mode connectivity. The work demonstrates how existing methods have to go through a valley in optimization during finetuning which degrades their immediate performance when starting the online phase. The work then proposes a score-matching loss to combat the issue. The method is evaluated on various standard offline to online control tasks.

**Compliance With Llm Reviewing Policy:**

Affirmed.

**Final Justification:**

One of my main concerns about objective mismatch has been clarified (namely that CalQL is in fact a MaxEnt method). I think the papers writing could be improved quite a bit but I don't think that should stop us from accepting a paper with decent results and some good insights. I have adjusted my score accordingly.

**Key Questions For Authors:**

Can you provide an explanation as to why you believe your approach is better at handling the offline to online transfer?

**Limitations:**

Yes

**Strengths And Weaknesses:**

# **Strengths**
**Motivation**
* The work is well motivated studying an interesting problem in the offline to online RL literature and it is quite timely.

**Clarity**
* The Figures are easy to read and articulate the results clearly

**Novelty**
* I am not aware of any work that has attributed the drop in performance when switching to fine-tuning in RL to mode connectivity and I believe it is a potentially interesting insight.

**Related Work**
* I’m not an expert in this particular part of the RL literature but most work that I am familiar with but the work seems to cite a decent amount of related other papers to contextualize itself.

**Empirical Design, Claims and Evidence**
* The experimental evaluation is quite extensive stretching across multiple methods and various environments providing evidence for the hypothesis that mode connectivity is a good explanatory signal.
* The choices of metrics are appropriate such as the usage of regret to characterize the unreliability of other approaches.

# **Weaknesses**
**Clarity**
* One of the biggest issues with this work is it’s lack of clarity. In particular the text is relatively difficult to read and requires the reader to jump back and forth a lot due to lack of structure. Some examples:
  * The text talks about SMAC quite a bit and even talks about results before ever introducing the approach. The work will refer to things without defining them and then define them two paragraphs later (e.g. the regret definition) which again makes the reader go back and fourth in the text.
  * The text describes the experimental setup just to then not talk about experiments but rather describe the approach. After reading the approach I then had to go back to understand the setup. Structuring things with a coherent flow would drastically improve the readability of the manuscript.
  * The text describes the method and inside the method will refer to future experiments instead of describing the results in the experiment section.
* The experiments are unstructured and not associated properly with claims, it’s just describing one figure after another without making it clear why this figure is being looked at now. Structuring experiments by questions or hypotheses that are being addressed would improve the readability of the experimental section.
* There are various explanations in text that rely on plots that are only provided in the Appendix. While I understand that space is limited, I think a clearer focus on what the work wants to show and then the inclusion of key experiments in the main text would improve the manuscript. Any experiment that is relevant to a key claim should be in the main text. Additional evidence can be moved to the Appendix but should not be treated as main evidence and rather supporting claims that already exist. One example is the whole discussion around AWR.

**Method design**
* The text states that the objective that is being optimized is theory-inspired but I am not following what the supposed theory contributes and how it might explain why the proposed approach works.
* I found the introduction of the Muon optimizer a bit counter-productive. It is unclear to me why it would be needed to support the claims the paper is trying to make. In fact, I looked at Appendix I, Figure 15 and to me it seems that it does not in fact contribute to fewer “dips” when switching from online to offline learning even though the text claims it does. In line 324L the text states Muon “ improved the offline-to-online transfer of SMAC”. That does not necessarily seem true. It seems to lead to faster convergence during online training on one task but is not needed to mitigate the offline-to-online drop in performance.. In my opinion this takes away from the story and only leads to more confounding factors and the explanation for the inclusion of Muon as a key contribution seems very hand-wavey.

**Empirical Design, Claims and Evidence**
* For most experiments, the paper does not report how many seeds are being used. While I am willing to believe that the results are significant because of the large differences in Regret, this number must be reported. I highly recommend a structured description of the setup at the beginning of the experimental section that includes a description of the statistical setup, the hypotheses, and the formal metrics.
* I think the experiments do not properly explain with evidence why the presented approach does not exhibit the linear failure mode case that others do. For example, in the experiments, one thing that I found a bit confusing is the focus on SAC as an online method. I think this introduces a confounding factor that is not properly ablated, namely the objective mismatch that is briefly discussed at the end of section 7. It is unclear to me how much of the effects that are observed in for instance Figure 2 stem from an objective mismatch between pre-training and fine-tuning. In particular, when moving to online learning with SAC, one introduces a term that changes the policy landscape with the entropy maximizer. None of the other approaches consider this term during pre-training but SMAC does. If we only look at TD learning, it seems that going from pretraining with TD3-BC or CalQL to finetuning with TD3-BC there are in fact linear connections (at least in Figure 2 and multiple tasks in Figure 6) between the solutions. As a consequence, I am not sure the paper does the best job it can in explaining where the linear connectivity exactly comes from. One potential experiment to run here is pre-training with a SAC style objective and without score matching.

# Neutral
* The paper claims that fixing the problem of the initial drop will let us use sample efficient online algorithms. However, the work then does not leverage recent advances on sample efficient learning. It feels to me that this would be an easy thing to test by using a more modern state-of-the art online approach.

# Summary:
I think this paper makes a very interesting connection between local mode connectivity and potential online transfer in RL. However, I think that the structure and writing of the manuscript could be improved drastically. That being said, the writing is sufficient for me to understand the results. The experimental evaluation demonstrating the approach's warm-start ability is convincing. However, the work does not provide a lot of explanation as to *why* the approach is working. In summary, I think the paper makes a decent contribution even though I believe that it could be significantly stronger with some minor extra work. I am not going to argue on behalf of the paper but I am also not going to be the one stopping it from getting accepted. Rather I am going to lean towards erring on the side of optimism and recommend weak acceptance.

---

> ### Author Rebuttal · Authors · 2026-03-31
>
> We appreciate the thoughtful review and level of detail shown by the reviewer. We're glad that the reviewer found our paper to be well motivated, providing an interesting insight into a timely problem, and that our experiments were extensive and convincing.
> # Clarity
> We commit to the following structural revisions: (1) add a 3-4 sentence SMAC preview immediately after the problem statement, covering the key identity $\nabla_a \log \pi^* (a|s) \propto \nabla_a Q^* (s, a)$, the regularizer, and its connection to smooth transfer; (2) absorb the offline-to-online setup (Section 4.2) into the problem statement and move benchmark details to the experiments section; (3) make the method section self-contained by removing forward pointers to experiments and justifying design choices inline; (4) reorganize the experiments around explicit research questions (smooth transfer, connectivity, sharpness/batch-size effects) with evidence presented alongside each; (5) move the AWR discussion to the appendix. All forward references will be accompanied by an inline definition.
> # Method Design
> > The text states that the objective...
>
> The theoretical inspiration for SMAC is the exact Max-Entropy identity: the optimal policy $\pi^* $ under SAC's objective satisfies $\nabla_a \log \pi^* (a|s) = \frac{1}{\alpha}\nabla_a Q^* (s,a)$ (Haarnoja et al., 2017). In Appendix P, we show that the same is true for PPO at the optimal policy. This identifies what property the Q-function needs to satisfy for the offline solution to be compatible with SAC's online solution.
>
> >I found the introduction of the Muon...
>
> Appendix I (Fig. 15) shows that replacing Muon with Adam causes performance drops in 3/6 environments for SMAC, while adding Muon to baselines gives no benefit. The mechanism we propose is that Muon takes gradient steps under the spectral norm (Bernstein & Newhouse, 2024), converging toward flatter regions of the Q-function landscape. Liu et al. (2023) show that flatter pre-training solutions transfer better downstream in supervised learning; we observe the analogous effect here.
> # Empirical Design, Methods, and Claims
> >For most experiments, the paper..
>
> All experiments use 5 seeds per offline-online pair per environment (4 for the Figure 3 connectivity experiment, as noted in the caption). We will add an explicit experimental setup section listing hypotheses, metrics, and settings.
>
> >I think the experiments...
>
> Establishing properties of neural network optima across training regimes is difficult: even a single width-512 layer admits hundreds of thousands of possible connecting directions. We therefore provide evidence from multiple angles: empirical verification (App. F), formal results (App. P), toy examples (Sec. 6), and cases where SMAC optima are also SAC optima (Sec. 6). These support that SMAC solutions lie near the optimization trajectory of online methods.
>
> SAC underlies most SOTA online methods (CrossQ, SIMBA, RLPD, REDQ), which differ mainly in update-to-data ratio, architecture, or parallelization. We also test TD3, TD3+BC, and AWR, showing that the result is not specific to SAC.
>
> CalQL already includes entropy maximization in its official implementation. Pure offline SAC diverges without regularization, a well-known failure mode, making CalQL the natural SAC-style baseline. SMAC also transfers smoothly to TD3, which has no entropy term.
>
> TD3+BC fine-tuning largely continues the offline objective: AWR (used by IQL) solves the constrained TD3 problem, while CalQL’s pessimism transfers into BC regularization; both penalize OOD actions. Smooth transfer therefore tracks objective compatibility: it holds in Kitchen but fails in pen, door, relocate, hopper, and walker2d, where TD3+BC reinforces poor demonstrations from the data or early rollouts. This supports our claim that connectivity depends on an objective match.
>
> # Key Question
> Section 6, Appendix P, and Appendix F provide motivation, theory, and empirical evidence that online actor-critic training implicitly pushes toward $\nabla_a \log \pi(a\mid s) \propto \nabla_a Q(s, a)$. By explicitly regularizing this property offline, SMAC learns actor-critics that are structurally closer to online-trained ones, reducing the mismatch between offline and online stable points. Using $\nabla_a \log \pi(a\mid s)$ as a proxy for $\nabla_a Q(s,a)$ therefore biases the critic and the induced actor toward an online-compatible solution, which we argue underlies SMAC's roughly monotonic improvement during value-based fine-tuning.
>
> This is why we argue that SMAC's offline checkpoint lies in a region of parameter space compatible with online fine-tuning: it explicitly enforces a property that SAC regularizes implicitly. We believe the same intuition explains transfer to TD3, as the $\alpha \to 0$ limit of max-entropy RL, and to TD3+BC and AWR, whose objectives can be expressed through forward or backward KL divergences where the same score-matching relation appears.

---

> > ### Author Rebuttal · Reviewer_2fpa · 2026-04-03
> >
> > Dear authors, I appreciate the clarifications.
> >
> > I was unaware that CalQL uses an entropy objective during pre-training which mitigates my concern around mismatch. I verified by double checking the code base. It seems that the paper argues that this linear mode connectivity phenomenon is exactly because of the mismatch. I would explicitly highlight this for CalQL in the paper.
> >
> > I think this point needs to be worked out more clearly in the text. It explains in part why in Figure 2 there is a much clearer connection between CalQL and SAC-finetuning. The focus should then not necessarily be general purpose matching with this objective but rather finding a pre-training objective that works well with approaches that learn a Boltzman distribution online.
> >
> > Some open points:
> > 1. I'm still not convinced I understand the reason this method works. The idea seems to derive form the Boltzman distribution of MaxEnt RL but then the approach still seems to work when fine-tuning with a method that is not a MaxEnt approach.
> > 2. My point was not that Muon performs worse or not, the claims are about quick transfer and in 5 out of 6 that is still true. One could remove the Muon optimizer part and make the exact same claims as the paper is currently making about transfer. Muon helping on some tasks is an interesting additional observation but it is not clear to me how it relates to the core finding of low-loss paths. My point is that it is *not* needed to get rid of the performance dip right after switching from offline to online RL that the paper argues comes from the missing linear path in the parameter landscape. I think it would be benefitial to describe why Muon would have anything to do with the mode connectivity.
> >
> > Overall, I think the results of the paper are promising and the mode-connectivity insight is interesting. Given that my main concern has been addressed I will adjust my score accordingly. However, I still think this paper could be significantly better with a clearer explanation as to why the approach works. It seems that other reviewers had similar concerns.

---

> > > ### Author Response · Authors · 2026-04-08
> > >
> > > Dear Reviewer,
> > >
> > > Thank you for the thoughtful reply and original review. We're glad that the
> > > mode-connectivity observation has been well received and that your main concerns
> > > have been addressed. We will try our best to clarify the two remaining open points.
> > >
> > > **1)** The idea does derive from the Boltzmann distribution and MaxEnt RL, but
> > > we want to offer a cleaner framing for why SMAC transfers to non-MaxEnt RL
> > > methods. By enforcing $\nabla_a \log \pi(a|s) \propto \nabla_a Q(s,a)$ during
> > > offline pre-training, SMAC shapes the Q-function's action-gradient landscape to
> > > be compatible with future online actor updates. For TD3 specifically, the
> > > deterministic policy gradient ascends $\nabla_a Q(s,a)$ directly at
> > > $a = \pi(s)$; a Q-function pre-shaped by score-matching is therefore already
> > > oriented in the direction TD3 will push the actor, with no MaxEnt assumption
> > > required. The smooth transfer curves to TD3 serve as the most direct empirical
> > > support for this. We will add this framing explicitly in Section 6.
> > >
> > > **2)** The introduction of Muon for SMAC is irrelevant to the claim about mode
> > > connectivity and offline-to-online transfer; however, it is directly relevant to
> > > the claim that SMAC achieves smooth offline-to-online transfer. Here, we want to
> > > respectfully flag what we believe is a misreading of Figure 15 in Appendix I,
> > > which shows that SMAC without Muon experiences worse performance at the first online checkpoint when compared to the checkpoint before any online learning in the Door, Walker, and Relocate environments.
> > >
> > > The connection to mode connectivity is the following: Muon manufactures gradient
> > > updates with a bounded spectral norm, biasing solutions toward flatter regions of
> > > the loss landscape. Flatter minima correspond to wider basins, meaning there are
> > > fewer high-loss barriers immediately around the parameterization found by SMAC.
> > > We believe this, in turn, makes it easier for an optimizer to move towards a
> > > linearly connected online solution, since fewer barriers exist near the current
> > > parameters.

---

### Decision · Program_Chairs · 2026-04-30

**Decision:**

Accept (regular)

**Comment:**

This paper addresses the critical challenge in offline RL: the immediate performance degradation that occurs when offline-trained actor-critics are fine-tuned online. The authors hypothesize that this drop is caused by the separation of offline and online optima within the loss landscape, which are separated by low-performance valleys. To bridge this gap, they introduce Score Matched Actor-Critic (SMAC). SMAC regularizes the $Q$-function during the offline phase by enforcing a first-order derivative equality between the policy's score and the action-gradient of the $Q$-function. This mechanism ensures that the learned offline maxima are connected to better online maxima via paths exhibiting monotonically increasing reward, thereby enabling smooth transfer to online algorithms such as Soft Actor-Critic and TD3.

The reviewers unanimously agree that this paper provides valuable insight into the dynamics of offline-to-online RL algorithms. The demonstration of mode connectivity is considered thorough and rigorous. Following the rebuttal, most initial concerns have been addressed, and the methods presented here hold potential as templates for developing more robust offline RL algorithms capable of transferring to general online fine-tuning methods. However, several outstanding issues should be addressed in the camera-ready version to fully realize the paper's potential. Specifically:
- **Scope of Applicability:** The regularization appears specifically tailored for entropy-regularized methods like SAC. It is unclear whether this method is theoretically guaranteed to work for other types of online fine-tuning methods. Furthermore, the claim that this stems from a theoretical property of the optimal policy in SAC may be overstated due to the approximations inherent in the behavior policy and dataset.
- **Training of Diffusion Models:** The sensitivity and optimal tuning of the score-matching regularization parameter are critical for successful transfer, and the paper must elaborate on the hyperparameter tuning process. The comparison of offline training performance against baselines (e.g., 3x gradient steps for IQL vs. 2x for CalQL) is noted in the rebuttal. However, a more honest comparison would involve directly showing the computational complexity for each method until convergence, as baseline methods cannot utilize the additional gradient steps afforded by SMAC. This comparison should be added to the final paper.
- **Generalizability of Connected Maxima:** While successful on several D4RL tasks, the concept of "connected maxima" is currently limited. Broader validation across different RL problem structures and environments is necessary for future study.

In summary, I find this paper to be interesting and highly valuable to the offline-to-online RL community. While the reviewers unanimously recommend acceptance, the authors must address these outstanding points in their final version. Congratulations on this significant work!